

# Pollen-based climatic reconstructions for the interglacial analogues of MIS 1 (MIS 19, 11 and 5) in the Southwestern Mediterranean: insights from ODP Site 976

Dael Sassoon*[1,4], Nathalie Combourieu-Nebout[1], Odile Peyron[2], Adele Bertini[3], Francesco Toti[3], Vincent Lebreton[1], Marie-Hélène Moncel[1]

*corresponding author:
Dr Dael Sassoon (dael.sassoon@gmail.com)

**Affiliations:**
1: UMR 7194, Histoire Naturelle de l´Homme Préhistorique, CNRS-MNHN, Institut de Paléontologie Humaine, Paris, France
2: Institut des Sciences de l'Évolution de Montpellier, UMR CNRS 5554 ISEM, Université de Montpellier, France
3: Dipartimento di Scienze della Terra, Universita di Firenze, Italy
4: Geosciences Barcelona (GEO3BCN), CSIC, Lluìs Solè i Sabarìs s/n, 08028 Barcelona, Spain

**Abstract**

Pleistocene interglacials, specifically MIS 19, 11 and 5, have been suggested as analogues of MIS 1 due to similar solar forcing patterns, greenhouse gas concentrations and sea levels. There has been substantial debate regarding which of these is the most suitable analogue and so far there has been no consensus, although what really emerges from recent work is the high variation in regional climate during these periods. One of the limiting factors in our understanding of these potential analogues is the fact that very few long-sequences cover the entire duration of these interglacials at high resolution.

In this study, a multi-method approach is used to produce climatic reconstructions for MIS 19, 11, 5 and 1, using pollen data derived from a single long marine core from ODP Site 976. This represents the first study which attempts to use pollen-based climatic reconstructions to compare MIS 1 with its analogues, representing a necessary contribution to the debate with a focus on the relationships between vegetation and climate in the southwestern Mediterranean.

Three methods of quantitative climate reconstruction have been adopted: the more widely used methods Modern Analogues Technique (MAT) and Weighted Average Partial Least Squares regression (WA-PLS), and a more recent machine-learning method known as Boosted Regression Trees (BRT). The reconstructions show consistent changes in temperature and precipitation during MIS 19, 11, 5 and 1, which correlate well with climatic changes observed in other regional and global proxies, and highlight distinct climatic characteristics of each interglacial period in the southwestern Mediterranean. MIS 19 exhibits high variability and colder temperatures compared to subsequent interglacials and the MIS 1. Conversely, MIS 11 displays warmer temperatures and greater stability, which makes it a useful analogue to understand prolonged interglacials, crucial considering the anthropogenic impacts on the duration of warm climate during the Holocene. MIS 5 exhibits overall warmer conditions, and its higher temperature coupled with fluctuations in solar forcing makes it a less suitable MIS 1 analogue.

Although past interglacials do not offer direct predictions for the Holocene's future, they provide essential insights into Earth's responses to various forcing factors, serving as crucial benchmarks for understanding the Mediterranean's sensitivity to global changes.

## 1. Introduction

The interglacials of the Pleistocene, particularly those of the past 1 Ma (1 million years) and specifically MIS 19 (ca. 795–755 ka BP), MIS 11 (ca. 424–365 ka BP) and MIS 5 (ca. 127–78 ka BP), have been source of increasing attention over the past two decades because several of them have been suggested as analogues of the Holocene (e.g. Loutre and Berger, 2003; McManus *et al.*, 2003; Tzedakis, 2010; Candy *et al.*, 2014; Yin and Berger, 2015; Giaccio *et al.*, 2015; Varvus *et al.*, 2018). Studying past interglacials can provide a framework to better evaluate the natural timing and duration of the Holocene, and examining the amplitudes and rates of climatic variability can give an indication of how the current interglacial may have been without anthropogenic interference, and how it could evolve under the presence of humans (Loutre and Berger, 2003; Candy *et al.*, 2014; Giaccio *et al.*, 2015). Furthermore, studying past interglacials may help understand abrupt climate change and the impact of these events on ecosystems and human populations (Loutre and Berger, 2003; Nomade *et al.*, 2019).

The selection of the interglacials MIS 19, 11 and 5 is mainly based on their similarities with MIS 1 in terms of astronomical configurations and greenhouse gas (GHG) concentrations (Yin and Berger, 2015). These



interglacials are characterised by low eccentricity and similar precession patterns to MIS 1, small variation in
insolation amplitudes, and elevated GHGs. However, the search for the best analogue has been source of constant
debate (Candy *et al*., 2014). Chiefly, the arguments have revolved around (1) the best alignment of the insolation
patterns between each interglacial and MIS 1, and (2) the structure and duration of these interglacials compared
with the Holocene (Candy *et al*., 2014; Past Interglacials Working Group of PAGES, 2016).
MIS 5, specifically substage 5e (ca. 128–116 ka BP)—known as the Eemian (Kukla *et al*., 1997)—has been
previously considered as a modern analogue due to the high temperatures over most of the Northern Hemisphere
(NH) and reduced ice sheets (Yin and Berger, 2015). However, the appropriateness of this interglacial was put in
question by Loutre and Berger (2003) due to its disproportionally high-amplitude changes in insolation and
shorter-lasting high $CO_2$ concentrations compared to the Holocene.
Rather, Loutre and Berger (2003) considered MIS 11 to be closer to MIS 1. Specifically, the climatic optimum
of MIS 11c (ca. 427–400 ka BP) has long been recognised as an analogue of the Holocene, owing to similar sea
levels, elevated temperatures, reduced astronomical forcing and high atmospheric $CO_2$ concentrations (McManus
*et al*., 2003; Desprat *et al*., 2005; Hes *et al*., 2022). This prolonged and stable period has received further attention
because it occurs after one of the harshest glacial conditions of the past 1 Ma (Berger and Loutre, 2003; Raymo
and Mitrovica, 2012; Oliveira *et al*., 2016), which had important implications on the rise of early hominin
populations including the spread of Neanderthals and their traditions across Europe and the Mediterranean
(Moncel *et al*., 2016; Blain *et al*., 2021; Sassoon *et al*., 2023). The suitability of MIS 11c as an analogue has been
supported by several studies (e.g. Berger and Loutre, 2002, 2003; McManus *et al*., 2003; Olson and Hearty, 2009;
Raymo and Mitrovica, 2012). Candy *et al*. (2014) pointed out that this interglacial matches the pattern of solar
insolation of the Holocene more closely than any other interglacial of the past 500 ka. However, recent studies
have questioned its reliability as analogue due to the unique antiphasing between precession and insolation and
obliquity—two precession peaks occurring during one obliquity cycle (Ruddiman, 2007; Tzedakis, 2010; Nomade
*et al*. 2019; Tzedakis *et al*., 2022).
Other authors argue that MIS 19 has greater resemblance to the Holocene, owing to a closer phasing of
obliquity and precession whereby the maximum obliquity is in phase with the minimum precession at the onset
of both interglacials (Tzedakis, 2010). This has been supported by several records in the North Atlantic and
Mediterranean (Pol *et al*., 2010; Tzedakis *et al*., 2012; Sanchez Goñi *et al*., 2016; Giaccio *et al*., 2015; Nomade
*et al*., 2019). This feature, along with similar duration of the climatic optimum, similar mid-June insolation and
comparably elevated $CO_2$ concentrations, has highlighted the viability of MIS 19 as a modern analogue. However,
Tzedakis (2010) demonstrated important differences between the trends of GHG concentrations during MIS 19
and MIS 1, and the climatic structure of MIS 19. Furthermore, it was found that MIS 19c was generally colder
than MIS 5e and MIS 11c (Jouzel *et al*., 2007), and therefore possibly less climatically comparable to the Holocene
especially in the Northern Hemisphere.
So far, there has been no consensus on which of these interglacials is the best MIS 1 analogue, and what really
emerges from the literature is the high variation in regional climate during MIS 19, 11 and 5. For instance,
the appropriateness of MIS 11 as an analogue was supported by McManus *et al*. (2003) in the North Atlantic and
by Wang *et al*. (2023) in China, but it was found to be questionable in the Nordic Seas in the study by Bauch *et
al*. (2000). This heterogeneity and lack of long cores makes it extremely important to compare these analogues
with MIS 1 at a regional scale, using high-resolution records with timeframes that encapsulate the entire
interglacials.
One region which can help shed some light on this debate is the Mediterranean, due to its high sensitivity to
climate change (Lionello and Scarascia, 2018). It is also an area which has been historically affected by
anthropogenic pressures, and is likely to be impacted by future warming and drought (Guiot and Cramer, 2016;
MedECC 2020; IPCC, 2022), making it imperative to understand the drivers of environmental and climate change
across the basin so that we can develop a better framework to predict the trajectory of our current interglacial
(Combourieu-Nebout *et al*., 2015). Moreover, several long cores are available for the Mediterranean region, such
as the terrestrial records from Tenaghi Philippon (Pross *et al*., 2015; Koutsodendris *et al*., 2023), Lake Ohrid
(Sadori *et al*., 2016; Wagner *et al*., 2019; Donders *et al*., 2021), Padul (Ortiz *et al* 2010; Camuera *et al*., 2018) and
marine records from the Iberian Margin (e.g. Sanchez Goñi *et al*., 2016). Some of these long pollen sequences
allowed to quantitatively reconstruct past climate changes during MIS 11 (Kousis *et al*., 2018), MIS 5 (Sinopoli
*et al*., 2019) and MIS 1 (Peyron *et al*., 2011, Camuera *et al* 2021).
Recent palynological studies from ODP Site 976 in the Alboran Sea, southwestern Mediterranean, have
yielded high-resolution pollen records for MIS 1 (Combourieu-Nebout *et al*., 2009, 2013; Dormoy *et al*., 2009),
MIS 5 (Masson-Delmotte *et al*., 2005), MIS 11 (Sassoon *et al*., 2023) and MIS 19 (Toti *et al*., 2020), providing a
unique opportunity to investigate the regional suitability of these interglacials as analogues of MIS 1 using proxies
from a single core. This study aims to provide quantitative estimates of past climate changes for each interglacial
by implementing a robust multi-method approach (Peyron *et al*., 2017; Salonen *et al*., 2019; Robles *et al*., 2023),
using pollen data derived from the long marine core of ODP Site 976. This approach enables a comparison of



Holocene analogues and represents a necessary contribution to the debate on the links between vegetation and
climate in the Mediterranean.
The objectives of this study are to:
1. Reconstruct temperature and precipitation parameters during MIS 19, MIS 11, MIS 5 and MIS 1 using
121        a pollen-based multi-method approach
2. Assess the reliability of multiple quantitative reconstruction methods
3. Compare climatic variability during the interglacials with local and global proxies
4. Evaluate the suitability of the interglacials as analogues of the Holocene in the Southwestern
125        Mediterranean.

## 2. Site description

This study used pollen records derived from the marine core of the Ocean Drilling Program (ODP) Site 976 in the
Western Alboran Sea (36°12.3'N 4°18.8'W), collected in 1999 during leg 161 (Shipboard Scientific Party, 1996).
This site (Fig. 1) is located about 110 km off the coast of the Strait of Gibraltar at a depth of 1108 m(Combourieu-
Nebout *et al.*, 1999, 2009; Gonzalez-Donoso *et al.*, 2000). Due to its susceptibility to polar, tropical, and Atlantic
influences, the Alboran Sea is extremely sensitive to climate changes on centennial and millennial scales, making
it an ideal location to study climatic variability and interglacial comparisons Alonso *et al.*, 1999; Combourieu-
Nebout *et al.*, 1999, 2002, 2009; Fletcher and Sanchez Goñi, 2008; Dormoy *et al.*, 2009; Toti *et al.*, 2020; Bulian
*et al.*, 2022).
The Alboran Sea measures 150 km in width and 350 km in length, forming a narrow extensional basin (Alonso
*et al.*, 1999) between the Mediterranean Sea to the east and the Atlantic Ocean to the west (Bulian *et al.*, 2022).
The northern coast of the basin borders with Spain while it borders with Morocco to the south. The Alboran Sea
is dominated by water circulation which is predominantly a result of the exchange of waters at the Strait of
Gibraltar whereby low-salinity waters from the Atlantic enter the basin and high-salinity waters from the
Mediterranean outflow into the ocean (Bulian *et al.*, 2022). This results in the Eastern Alboran Gyres (EAG) and
the Western Alboran Gyres (WAG) (Bulian *et al.*, 2022), two anti-cyclonic gyres (fig. 1).
This part of the Mediterranean is affected by the Southern Azores cyclone resulting in long, dry summers with
mean temperatures typically exceeding 20°C. In contrast, winters are mild and rainy, with temperatures ranging
10°C on the coast and -7°C at higher elevations resulting in an altitudinal gradient; annual precipitation is usually
400–1400 mm (Quézel and Médail, 2003; Grieser *et al*., 2006).

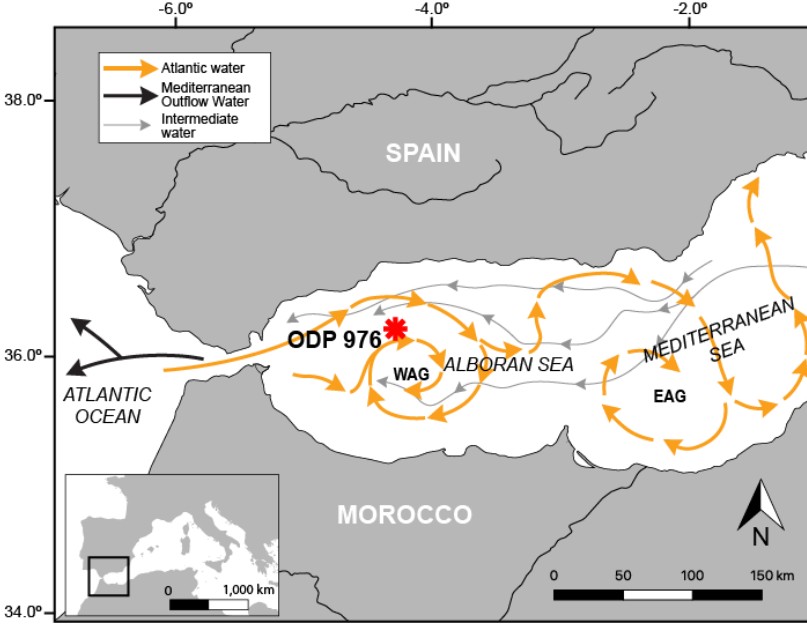

**Figure 1 - Map showing the location of ODP Site 976 and the present-day surface and water circulation in**
**the Alboran Sea (modified from Combourieu-Nebout et al., 1999).**





Vegetation cover is a function of an altitudinal gradient owing to the presence of the Moroccan Rif and Betic Cordillera (Quézel and Medail, 2003). The coast is dominated mainly by steppe with *Lygeum*, *Artemisia* and Mediterranean taxa (e.g. *Olea*, *Phillyrea, Pistacia*, and *Quercus ilex*). Humid-temperate oak forest with *Quercus deciduous* and Ericaceae dominates the mid-altitudes. Higher elevations are mainly characterised by cold-temperate coniferous forests with *Pinus* and *Abies*. Although once more spread in the Mediterranean, *Cedrus* is only found now at higher elevations in Morocco (Ozenda, 1975; Rivas Martinez, 1982; Barbero *et al.*, 1981; Benabid, 1982).

### 3. Methods

3.1 Fossil pollen datasets

The fossil pollen datasets used to run the pollen-based quantitative climatic reconstructions are all obtained from the ODP Site 976 marine record from the studies listed below. All records excluded *Pinus*, due to its overrepresentation in marine samples (Heusser and Balsam, 1977; Naughton *et al.*, 2007). The ages used in this study are in calendar ka (cal ka).

- The pollen record for MIS 19 (Toti *et al.*, 2020) comprises 102 samples. The chronology was based on the initial age models from de Kaenel *et al.* (1999) and Grafenstein *et al.* (1999). Samples were taken every 10 cm, yielding an average temporal resolution of 450 years between samples.
- The pollen record for MIS 11 has a total of 141 samples (Sassoon *et al.*, 2023). The chronology for the fossil pollen record is based on von Grafenstein *et al.* (1999). Age interpolation revealed a lowermost age of 433.868 ka BP at 118.8 m and an uppermost age of 356.456 ka BP at 98.85 m. The pollen record for MIS 11 has an almost consistent resolution of 10 cm, achieving average temporal resolutions of ca. 128 years between samples.
- The MIS 5 record has 105 samples (Combourieu-Nebout *et al.*, 2002; Masson-Delmotte *et al.*, 2005). The chronology for this record was based on the age model by Combourieu-Nebout *et al.* (2002) but has been extended to 130 ka BP by correlation with deep sea core MD95-2042 (Shackleton *et al.*, 2003) and NorthGRIP $\delta^{18}$O record (NorthGRIP, 2004). Samples were taken at an average resolution of 10 cm, yielding an average temporal resolution of 500 years between samples.
- The record for MIS 1 was based on the uppermost 10m of the ODP Site 976 core, with a total of 136 samples (Combourieu-Nebout *et al.*, 2009). The chronology is built on ten $^{14}$C AMS radiocarbon ages, specifically measured on monospecific samples of *Globigerina bulloïdes* and *Neogloboquadrina pachyderma*, which revealed a lowermost age of 25 cal ka. The pollen analysis involved sampling at 10 cm intervals, with a higher resolution of 1–5 cm for the Bølling/Allerød and the early Holocene, yielding a resolution which varies from ∼20–40 years during the abrupt events to 200–500 years elsewhere.

3.2 Pollen-based climate reconstructions methods

Three methods of climate reconstruction have been used to derive quantitatively changes in temperature and precipitation parameters for the ODP Site 976 pollen records: Modern Analogues Technique (MAT; Guiot, 1990), Weighted Average Partial Least Squares regression (WA-PLS; Ter Braak and Juggins, 1993) and Boosted Regression Trees (BRT; Salonen *et al.*, 2014).

MAT and WA-PLS have been previously used for climate reconstruction focusing on different time periods in the Mediterranean region on both terrestrial and marine pollen records (e.g. Cheddadi *et al.*, 1998; Davis *et al.*, 2003; Pross *et al.*, 2009; Peyron *et al.*, 2011, 2013, 2017; Joannin *et al.*, 2012; Kotthoff *et al.*, 2008; Dormoy *et al.*, 2009; Sanchez Goñi *et al.*, 2012; Desprat *et al.*, 2013; Sadori *et al.*, 2013; Mauri *et al.*, 2015; Kousis *et al.*, 2018; Ardenghi *et al.*, 2019; Sinopoli *et al.*, 2019; Koutsodendris *et al.*, 2019; Robles *et al.*, 2023; Herzschuh *et al.*, 2023). The results are often well-supported by other Mediterranean records and independent proxies such as alkenones and other biomarkers, $\delta^{18}$O isotopes and sea surface temperature reconstructions, showing the reliability of these methods.

MAT uses the present-day environment to quantitatively reconstruct past climate derived from fossil assemblages (Chevalier *et al.*, 2020). MAT functions by determining the degree of dissimilarity between past pollen assemblages and modern pollen data. By using squared-chord distance calculations, MAT selects a number of modern pollen data considered as analogues for each fossil pollen assemblage to infer past climatic values (Guiot, 1990).

In contrast to the MAT which is an "assemblages approach", the WA-PLS method is a true transfer function meaning that it requires statistical calibration between the climate parameters and modern pollen assemblages (Chevalier *et al.*, 2020). It is a regression method which supposes the unimodal relationship between pollen percentages and climate parameters.

In comparison to the other methods, BRT is a machine learning method developed for ecology (De'ath, 2007; Elith *et al.*, 2008) and has recently been adopted for palaeoecology and palaeoclimatic reconstructions (Salonen *et al.*, 2014). It uses random binary splitting and cross-validation to predict the relationship between climatic



variables and pollen assemblages (Chevalier *et al*., 2020). In BRTs, great numbers of simple regression-tree
models are combined to produce a final model optimised for prediction, using cross-validation for model building.
This approach is promising for Mediterranean terrestrial records (Robles *et al*., 2023; d'Oliveira *et al*., 2023) but
has never been tested on marine pollen records or indeed records of the Mid-Pleistocene.
All three methods were calibrated using an updated version of the high-quality and taxonomically consistent
modern pollen dataset (Peyron *et al*., 2013; Dugerdil *et al*., 2021) containing 3,267 samples from European and
Mediterranean regions. *Pinus* has been omitted because its overrepresentation in the Mediterranean pollen
spectrum could mask climatically-related signals from other taxa (Sinopoli *et al*., 2019).
In this study, we reconstructed the following climatic parameters: (1) mean annual temperature (TANN); (2)
mean temperatures of the coldest month (Twin) and (3) warmest month (Tsum); (4) mean annual precipitation
(PANN); (5) summer precipitation (Psum); (6) winter precipitation (Pwin). The entire dataset includes the
parameters for growing degree days above 5°C (GDD5), the ratio of actual over potential evapotranspiration
(AET/PET), and further seasonal parameters including autumn and spring temperature and precipitation (Taut and
Tspr, Paut and Pspr, respectively). The studies by Combourieu-Nebout *et al*. (2009) and Dormoy *et al*. (2009),
which implement pollen-based reconstructions for MIS 1 using pollen data from ODP Site 976, represent a crucial
foundation for the present paper. While providing guidance, however, these previous studies only applied the
MAT method, therefore the application of new methods is necessary to enable the comparison with the results for
the other Holocene analogues.
Quantitative reconstruction methods and reliability tests were carried out with the software R using the
package '*rioja*' (Juggins, 2020). The reliability of pollen-inferred climate reconstruction methods was estimated
trough bootstrapping cross-validation by calculating the correlation coefficient values between the variables ($R^2$),
and using the Root Mean Square Error (RMSE) criterion. Higher $R^2$ and lower RMSE indicate greater validity of
the reconstructed parameters. Loess smoothing of 0.2 was applied to the raw data in the plots to view the overall
trends of the parameters.
**4. Results and discussion**
4.1 Multi-method approach: reliability and differences between the methods
The temperature and precipitation reconstructions for the three methods yielded coherent results for the
interglacials and interstadials investigated, aligning reasonably with trends observed in other regional climatic
proxies (section 4.2).
A comparison of the methods across the four interglacials, based on the $R^2$ and RMSE values, reveals
discrepancies in the performance trends. To exemplify these differences between methods, the $R^2$ and RMSE
results for TANN and PANN are shown in table 1. Overall, the models reconstruct TANN more consistently than
PANN, based on the significant difference between the RMSE values for these parameters across all MIS periods.
BRT consistently demonstrates robust performance, with high $R^2$ values ranging from 0.918 to 0.920 for TANN
and 0.822 to 0.826 for PANN, alongside low RMSE values compared to the other methods. The MAT method,
akin to BRT, shows strong performance with high $R^2$ values ranging from 0.865 to 0.866 for TANN and slightly
lower values of 0.711 to 0.713 for PANN, alongside comparatively low RMSE values. However, in comparison
to BRT, the MAT method tends to have slightly lower $R^2$ and higher RMSE, and there is a greater degree of
fluctuation for the parameters reconstructed which is interpreted as greater sensitivity to changes in the pollen
assemblages. In contrast, WA-PLS exhibits lower $R^2$ values (ranging from 0.445 to 0.683) and higher RMSE
values (ranging from 4.271 to 232.650) across both TANN and PANN parameters, indicating potentially poorer
model performance compared to BRT and MAT. Notably, BRT and MAT methods demonstrate greater
consistency in performance across interglacials and parameters compared to WA-PLS, suggesting their superior
efficacy in reconstructing climatic parameters across different temporal periods.
The observed trends in performance of the methods for TANN and PANN are applicable across all parameters
reconstructed (see supplementary data); BRT and MAT consistently exhibit strong performance characterized by
high $R^2$ values and low RMSE scores for all reconstructed parameter, while the WA-PLS method has lower $R^2$
values and higher RMSE scores across the board, suggesting a tendency toward less accurate reconstructions.





**Table 1 – R² and RMSE results from the methods BRT, WA-PLS and MAT for selected parameters (TANN and PANN) for the interglacials analysed in this study.**

| | | MIS 1 | | MIS 5 | | MIS 11 | | MIS 19 | |
|---|---|---|---|---|---|---|---|---|---|
| | | $R^2$ | RMSE | $R^2$ | RMSE | $R^2$ | RMSE | $R^2$ | RMSE |
| *BRT* | TANN | 0.918 | 2.965 | 0.919 | 2.960 | 0.920 | 2.962 | 0.919 | 2.947 |
| | PANN | 0.826 | 175.892 | 0.822 | 176.922 | 0.825 | 176.590 | 0.823 | 176.822 |
| *WA-PLS* | TANN | 0.683 | 4.271 | 0.683 | 4.275 | 0.683 | 4.277 | 0.683 | 4.275 |
| | PANN | 0.453 | 232.518 | 0.453 | 232.646 | 0.453 | 232.552 | 0.445 | 232.650 |
| *MAT* | TANN | 0.865 | 3.067 | 0.866 | 3.063 | 0.865 | 3.072 | 0.865 | 3.067 |
| | PANN | 0.713 | 184.261 | 0.712 | 184.385 | 0.711 | 187.333 | 0.711 | 183.010 |

4.2 Climatic reconstructions for each interglacial
*4.2.1 MIS 20–19 (803–748 ka BP)*
The reconstructions for MIS 20–19 show large-amplitude changes in temperature and precipitation (Fig. 2, Tab.
2). During the period reconstructed for the MIS 20 glacial between 803–786 ka BP, results indicate a cold and dry
climate, linked to the occurrence of steppic and semi-desertic taxa such as *Artemisia,* Amaranthaceae and
*Ephedra*, which are adapted to cold climates (Toti *et al*., 2020). Throughout MIS 20, TANN fluctuates around 4.7
ºC, with PANN averaging approximately 460 mm/yr, although there is a contrast between the periods 803–800
ka BP and 799–787 ka BP (Fig. 2, Tab. 2). In the former period, PANN is around 600 mm/yr and Pwin around
200 mm/yr, while in the latter period PANN decreases to below 400 mm/yr and Pwin to below 50mm/yr (Fig.
S2). The transition to harsher conditions during the late MIS 20 (around 799 ka BP) was associated with colder
conditions, as evidenced by palynological and foraminiferal records (Toti *et al*., 2020). This corresponds to a
shutdown of the Atlantic Meridional Overturning Circulation (AMOC) during that time (Cacho *et al*., 2000;
Moreno *et al*., 2004). Maiorano *et al*. (2016) observed this in the Montalbano Jonico section (southern Italy) and
referred to it as a Heinrich-type event (Med-HTIX) in analogy to those of the last termination (TI), and similarly
the warm-cold episodes during TIX have been named the Bølling-Allerød-like (Med-BATIX) and Younger-
Dryas-like (Med-YDTIX) events (Maiorano *et al*., 2016).
From 788–774 ka BP, the reconstructions for TANN indicate a rise from 2–7 ºC during the glacial to 6–13 ºC,
indicating the transition to MIS 19 (Fig. 2). This period is equivalent to the climatic optimum MIS 19c. This trend
is also indicated by PANN, which increases from 350–500 mm during the glacial to between 600–800 mm across
the three methods during the climatic optimum, indicating warmer and wetter conditions compared to MIS 20
(Toti *et al*., 2020). This climatic amelioration is interrupted by a short-lived event to cooler and drier conditions
and a change in seasonality around 785 ka BP. Twhis event has been observed in other pollen records including
Montalbano Jonico (Bertini *et al*., 2015) and speleothem records like Sulmona (Regattieri *et al*., 2019).
In the Alboran Sea, a peak in warmth and humidity is observed around 778 ka BP throughout the three
methods, although some differences in the methods are observed, where WA-PLS and BRT suggest a more
gradual temperature and precipitation increase than MAT, which indicates greater amplitude fluctuations (Fig. 2).
TANN averages between 5 and 10ºC and PANN is around 500–700 mm/yr, with Pwin values of around 150–300
mm/yr and Twin around 0ºC, suggesting temperate summers and mild winters during MIS 19c (Fig. 2, Fig. S2).
These reconstructions correlate well (Fig. 2) with the progressive increase in $CH_4$ and $CO_2$ observed in the EPICA
ice cores (Jouzel *et al.*, 2007; Nehrbass-Ahles *et al.*, 2020), and decline in Atlantic $\delta^{18}O$ (e.g. Voelker *et al.*, 2010;
Oliveira *et al.*, 2016).
There is a decisive fall in temperature centred between 774–771 ka BP, along with a slight decrease in
precipitation (Tab. 2), consistent with a return to colder and drier conditions during MIS 19b-a (Toti *et al*., 2020).
Twin fluctuates from -9°C to 7°C, indicating substantial variability in winter temperatures, while Tsum ranges
from 13°C to 22°C, suggesting fluctuations in summer warmth (Fig. S2). TANN varies between 0°C and 14°C,
indicating overall climatic changes throughout the year. PANN ranges from 370 mm to 750 mm, reflecting
fluctuations in annual precipitation levels. This is followed by three large-amplitude fluctuations during MIS 19a
(Fig. 2, Tab. 2), with extreme peaks at 770 and 765 ka BP, separated by two significant events of climatic
deterioration at 768 and 764 ka BP, which are linked to the high frequency alternation between forested and open
vegetation observed in the pollen record. This shows good agreement with oscillations in the benthic $\delta^{18}O$ record
of Montalbano Jonico from Nomade *et al*. (2019), who labelled these 19a-1, 19a-2 and 19a-3. These fluctuations
also correlate well with those observed in the benthic $\delta^{18}O$ record from Sulmona (Giaccio *et al*., 2015; Regattieri



*et al*., 2019), Atlantic δ[18]O (e.g. Voelker *et al.*, 2010; Oliveira *et al.*, 2016) as well as the CH$_4$ (Loulergue *et al*.,
2008) and CO$_2$ observed in the EPICA ice cores (Jouzel *et al.*, 2007; Nehrbass-Ahles *et al.*, 2020). These climatic
oscillations may have been caused by a shift in the position of the ITCZ causing northward pressure on the
Mediterranean leading to more arid summers and enhanced winter precipitation (Toti *et al*., 2020).

**Table 2 – Summary of results of the pollen-based climatic reconstructions for MIS 20–19**

| Interval | Age (ka BP) | Summary |
|---|---|---|
| **MIS 19a and 19b** | 773–756 | Decisive fall in temperature centred between 773–771 ka BP.<br>Slight decrease in precipitation but to a lesser extent and consistent with a return to colder and drier conditions.<br>Three large-amplitude fluctuations with extreme peaks at 770 and 765 ka BP, separated by two significant events of climatic deterioration at 768 and 764 ka BP.<br>Continued large-amplitude changes in temperature and precipitation. |
| **MIS 19c climatic optimum** | 786–773 | TANN shows a rise from 2–7 °C during the glacial to 6–13 °C.<br>PANN increases from a range of 350–500 mm during the glacial to between 600–800 mm.<br>MAT suggests the largest changes in both temperature and precipitation.<br>Peak in warmth and humidity observed synchronously around 778 ka BP. |
| **MIS 20/19 transition** | 803–786 | MAT suggests the largest changes in both temperature and precipitation during this transition.<br>Shift from glacial conditions (MIS 20) to interglacial conditions (MIS 19). |




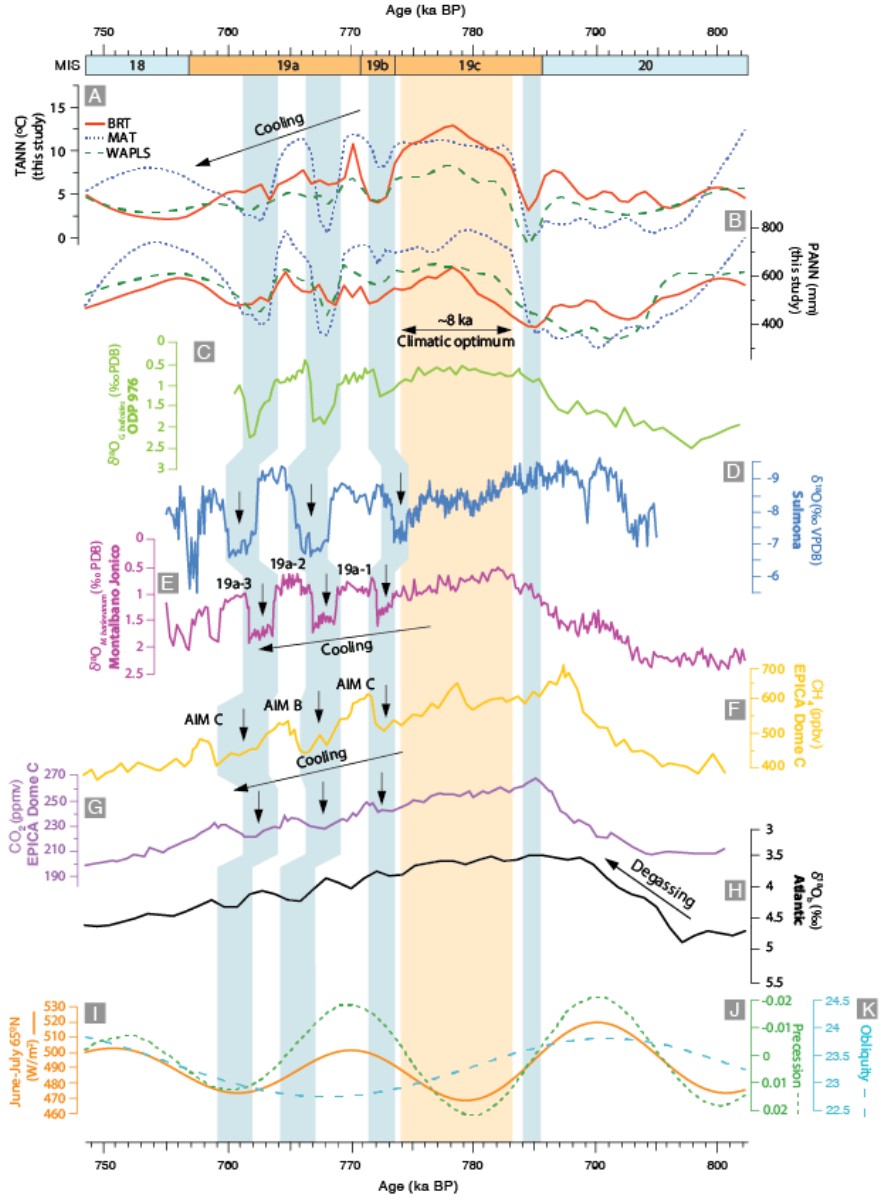

**Figure 2 – Comparison of the pollen-based quantitative reconstructions from ODP976 for MIS 19, (A)**
**TANN and (B) PANN (BRT=red solid line; MAT=blue dotted line; WA-PLS=green dashed line), with other**
**regional and global proxies: (C) $\delta^{18}O_{G.\ bulloides}$ record from ODP976 (Toti et al., 2020); (D) $\delta^{18}O$ records of**
**Sulmona basin sediments (Regattieri et al., 2019); (E) $\delta^{18}O_{M.\ barleeanum}$ record from Montalbano Jonico**
**(Nomade et al., 2019); (F) Methane ($CH^4$) atmospheric concentrations (Loulergue et al., 2008) and (G) $CO^2$**
**atmospheric concentrations from Antarctic EPICA Dome C ice cores (Nehrbass-Ahles et al., 2020); (H)**
**Atlantic $\delta^{18}O$ (Voelker et al., 2010); (I) Summer insolation (Laskar et al., 2004); (J) Precession index and**
**(K) Obliquity curve (Berger and Loutre, 1991). Orange band indicates the period encompassing the**
**climatic optimum, and the blue bands highlight major millennial-scale climatic events.**



*4.2.2 MIS 12–11 (434–356 ka BP)*
Between 434 and 427 ka BP, reconstructions for the end of MIS 12 show a generally cold and dry climate (Fig.
3). Annual temperature reconstructions reveal consistently low values across methods, with the coldest period
occurring before 430 ka BP (Tab. 3). During this period, Twin shows temperatures ranging -5–0 ℃ and Tsum
does not rise above 17 ℃ (Fig. S3). Following a brief warming around 430 ka BP, a rapid return to colder
conditions is observed at 428–426 ka BP across all three methods (Fig. 3). This abrupt shift to colder conditions
coincides with decreased sea surface temperatures (SSTs) and increased $\delta^{18}O_{G.\ bulloides}$ in the record from the same
ODP976 core by Brice (2007), who made the analogy with a Younger Dryas-like (YD-l) event. Other studies refer
to this as the Ht4 Heinrich-type event (Hodell *et al*., 2008; Rodrigues *et al*., 2011; Girone *et al*., 2013; Marino *et
al*., 2018). Vázquez Riveiros *et al*. (2013) noted enhanced Ice Rafted Debris (IRD) coeval with a sudden decrease
in North Atlantic SSTs during this event, indicating significant ice-rafting. Other pollen-based reconstructions,
particularly those from Lake Ohrid which used the MAT method (Kousis *et al*., 2018), show a short-lived decrease
in temperatures, precipitation, and forest cover prior to the onset of warmer and wetter conditions during
Termination V.
From 427 to 405 ka BP, a period with consistently high temperatures and precipitation are observed (Fig. 3,
Tab. 3), consistent with the warmest part of MIS 11, substage MIS 11c (Sassoon *et al*., 2023). This transition has
also been observed in other records (Fig. 3) in the Mediterranean region (Tzedakis, 2010; Girone *et al*., 2013;
Kousis *et al*., 2018; Koutsodendris *et al*., 2019; Ardenghi *et al*., 2019; Azibeiro *et al*., 2021), the North Atlantic
off the Iberian coast (Desprat *et al*., 2005; Oliveira *et al*., 2016) and continental Europe (Reille and de Beaulieu,
1995). TANN rises from around 8 ℃ to ~10–15 ℃, over the timeframe of ca. 2,000 years. BRT and WA-PLS
show Tsum values of around 18 ℃, while the MAT method estimates warmest-month temperatures of over 22 ℃
(Fig. S3). This warming is in agreement with the expansion of forest biomass observed in several other records
from across the Mediterranean basin throughout Termination V including Lake Ohrid (Kousis *et al*., 2018),
Tenaghi Philippon (Wijmstra and Smit, 1976; Tzedakis *et al*., 2006; Pross *et al*. 2015; Ardenghi *et al*., 2019;
Koutsodendris *et al*., 2023) and Bouchet/Praclaux (Reille and de Beaulieu, 1995). This increase in temperatures
during MIS 11c may be linked to the MIS 11.3 light isotopic event (Oliveira *et al*., 2016) and the highest summer
insolation recorded for MIS11 in the Northern Hemisphere (Sassoon *et al*., 2023). The warming trend is also
coeval with the rise in Antarctic air temperatures and Atlantic $CO_2$ records (Fig. 3) (Jouzel *et al*., 2007; Loulergue
*et al*., 2008; Nehrbass-Ahles *et al*., 2020). These results correlate with the highest SSTs, highest $CO_2$ and $CH_4$
concentrations (Jouzel *et al*., 2007; Nehrbass-Ahles *et al*., 2020), and reduced $\delta^{18}O$ (e.g. Voelker *et al*., 2010;
Oliveira *et al*., 2016).
Precipitation also increases during the climatic optimum, suggesting warm and humid conditions (Fig. 3, Tab.
3). Annual precipitation results from BRT and WA-PLS show a rise from 500 mm/yr during the glacial to 600
mm/yr for MIS 11c in the period between 429–427 ka BP, while MAT suggests a larger amplitude of change from
around 380 mm/yr to 600 mm/yr. These results are consistent with pollen-based quantitative reconstructions of
Kousis *et al*. (2018) at Lake Ohrid, which suggest a shift to more a humid and warmer climate at the beginning of
MIS 11c. However, the reconstructions for precipitation at Lake Ohrid are comparatively higher than the results
for ODP 976, evidenced by a rise in PANN to 800–1000 mm/yr at Lake Ohrid (Kousis *et al*., 2018). At Tenaghi
Philippon, precipitation reconstructions derived from calcium/iron (log(Ca/Fe)) ratio by Koutsodendris *et al*
(2023) show that MIS 11c was one of the wettest interglacials at this site with a significant difference between the
climatic optimum and the rest of MIS 11. This is a significant finding because this corroborates the hypothesis
put forward by several authors (Kandiano *et al*., 2012; Kousis *et al*., 2018; Sassoon *et al*., 2023) who suggested,
on the basis of pollen assemblages, that during MIS 11c, the climate in the southwestern Mediterranean was
warmer and drier than Lake Ohrid and Tenaghi Philippon in the Balkan Peninsula. Although this might be an
effect of a difference in altitude between the sites (which might also explain the difference in annual temperature)
and the nature of the substrates observed (marine vs. terrestrial), it might be indicative of an easterly humidity
gradient within the wider region owed to the formation of a bipolar see-saw pattern in precipitation between the
western and eastern Mediterranean possibly caused by a weakening of the AMOC during the deglaciation (Kousis
*et al*., 2018).
During the MIS 11c optimum, a noteworthy fluctuation occurs around 408 ka BP, mainly indicated in our
reconstructions by a decrease in PANN (Fig. 3). This is related to a moderate-intensity contraction in temperate
and Mediterranean forests (Sassoon *et al*., 2023). Oliveira *et al*. (2016) and Kousis *et al*. (2018) have linked this
forest contraction with the "Older Holstenian Oscillation" (OHO), also found in other records from Europe (West,
1956; Kelly, 1964; Turner, 1970; Kukla, 2003; Koutsodendris *et al*., 2011, 2012, 2023; Tye *et al*., 2016). Our
reconstructions indicate a reduction in TANN by about 1-2℃, and in PANN by 50 mm/yr on average across the
three methods. This appears to be less intense than the changes inferred for Lake Ohrid (Kousis *et al*., 2018) or
Tenaghi Philippon (Ardenghi *et al*., 2019), which suggest a higher amplitude of change in both precipitation and
temperature in the Balkans.





Between 400 and 356 ka BP, the substages MIS 11a and 11b exhibit reduced climate variability. Around 400–390 ka BP, a synchronous decline across the reconstructions for temperature and precipitation is interpreted as a cooler and drier phase, recognized as MIS 11b, connected to a decrease in summer insolation. The reconstructions show a decline in temperature and precipitation parameters centred around 398 ka BP (Fig. 3). Similarly, reconstructions for Lake Ohrid demonstrate reductions in TANN and PANN (Kousis *et al.*, 2018), indicating a synchronous cooling on land and the sea. Around 390–367 ka BP, recognised as substage MIS 11a, a return to warmer and more humid conditions, though relatively less temperate as he conditions during MIS 11c, are observed. Temperature reconstructions vary depending on methods, with WA-PLS and BRT indicating less variation than MAT suggests. PANN and Pwin also increase compared to previous levels at the end of MIS 11b, showing high variability during MIS 11a (Fig. 3, Fig. S3). Overall, however, these trends correlate with patterns observed in palaeoclimatic records from the North Atlantic and Mediterranean and seem to reflect summer insolation (Candy *et al.*, 2014, 2024).

The fluctuations during MIS 11a and 11b can be correlated with the light isotopic events 11.24, 11.23 and 11.22 (Fig.3), observed in $\delta^{18}O$ records (Brice, 2007; Desprat *et al.*, 2005; Oliveira *et al.*, 2016). Particularly, the drop in precipitation and temperature around 397 ka BP, reflective of the rise in steppe taxa in ODP 976, is synchronous with light isotopic event 11.24, also observed at IODP Site U1385 (Oliveira *et al.*, 2016), MD01-2447 (Desprat *et al.*, 2005, 2007), at Lake Ohrid (Kousis *et al.*, 2018), and at Tenaghi Philippon (Ardenghi *et al.,* 2019). The alkenone-based SST record from MD03-2699 show reductions to ~10ºC (Rodrigues *et al.*, 2011). This trend can also be compared with falls in $CO_2$ and $CH_4$ concentrations in the Antarctic EPICA records, which exemplify the sensitivity of the Mediterranean to global-scale climate change.

From 367 ka BP onwards, the temperature and precipitation reconstructions across all methods collectively suggest a transition to a significantly colder and drier climate, consistent with the beginning of the glacial inception of MIS 10.

**Table 3 - Summary of results of the pollen-based climatic reconstructions for MIS 12–11**

| Interval | Age (ka BP) | Summary |
|---|---|---|
| **MIS 11a and b** | 400–367 | Decline in TANN to around 10 ºC. Twin falls to a minimum of 0 ºC at 398 ka BP.<br>Tsum shows consistent decline to ~20 ºC at 397 ka BP.<br>Precipitation parameters for MIS 11b, display a fall in precipitation around 380 ka BP.<br>MAT and BRT suggest a progressive rise in precipitation from 400 ka BP culminating at 395 ka BP. |
| **MIS 11c climatic optimum** | 427–400 | Consistently high temperatures and precipitation.<br>TANN ranges between 10 and 15 ºC, indicating relative climatic stability.<br>Three distinctive temperature peaks observed, with the third around 405 ka BP. |
| **MIS 12/11 transition** | 433–427 | Lowest annual temperatures (~5 ºC) before 430 ka BP.<br>Brief temperature peak around 430 ka BP, followed by rapid return to cold conditions at 428 ka BP.<br>Decline in precipitation until 430 ka BP, PANN ranging 400–600 mm.<br>Transition to warmer, more humid climate around 428 ka BP with temperatures over 22 ºC and annual precipitation rising to 600 mm. |

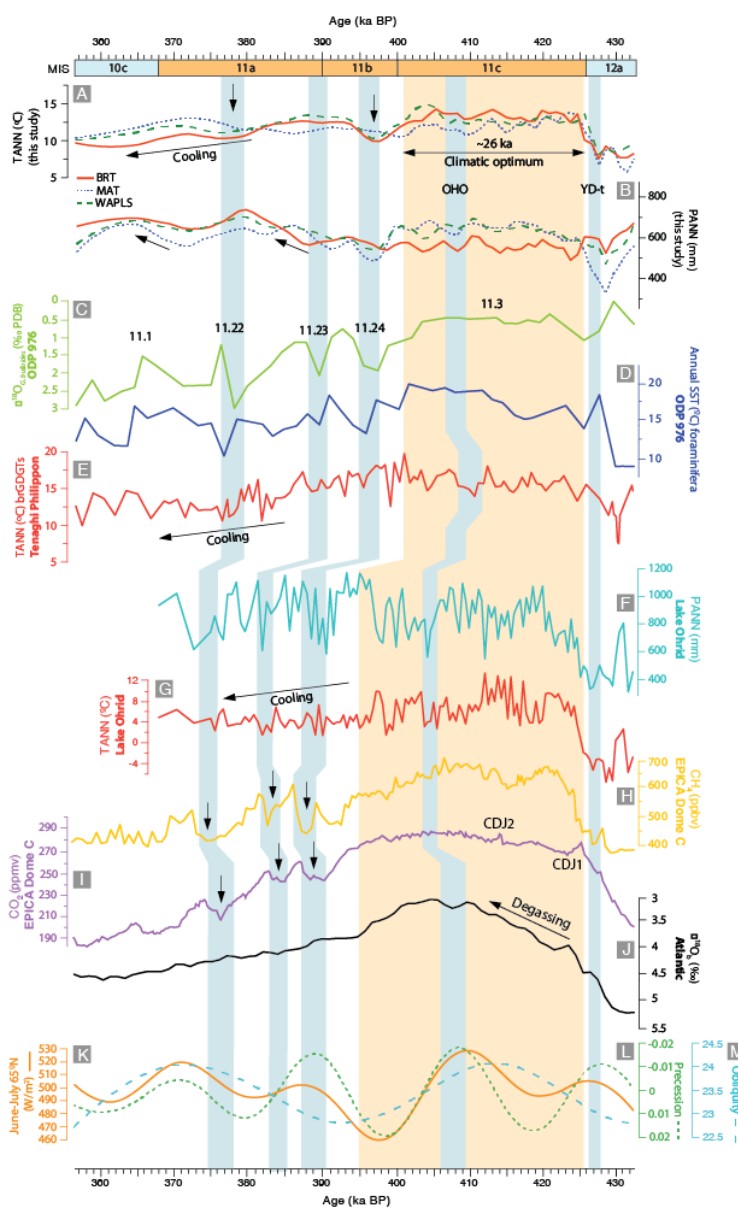

**Figure 3 – Comparison of the pollen-based quantitative reconstructions from ODP976 for MIS 11, (A)**
**TANN and (B) PANN (BRT=red solid line; MAT=blue dotted line; WA-PLS=green dashed line), with other**
**regional and global proxies: (C) δ¹⁸O$_{G. bulloides}$ and (D) annual SSTs from function transfer of foraminiferal**
**assemblages from ODP976 (Brice, 2007); (E) brGDGT-derived TANN from Tenaghi Philippon (Ardenghi**
**et al., 2019); (F) PANN and (G) TANN from Lake Ohrid derived through the MAT method (Kousis et al.,**
**2018); (H) Methane (CH₄) atmospheric concentrations (Loulergue et al., 2008) and (I) CO₂ atmospheric**
**concentrations from Antarctic EPICA Dome C ice cores (Nehrbass-Ahles et al., 2020); (J) Atlantic δ¹⁸O**
**(Voelker et al., 2010); (K) Summer insolation (Laskar et al., 2004); (L) Precession index and (M) Obliquity**
**curve (Berger and Loutre, 1991). Orange band indicates the period encompassing the climatic optimum,**
**and the blue bands highlight major millennial-scale climatic events.**



### 4.2.3 MIS 6–5 (133–80 ka BP)

The climatic reconstructions for the period between 133 and 128 ka BP, equivalent to the end of the MIS 6 glacial period, also referred to as the penultimate glacial, indicate cold and dry conditions, though some differences between methods are observed (Fig. 4). Generally, the three methods show low values of TANN (range of 10-13ºC) and Twin (range of -5ºC to 3ºC) for the glacial period (Fig. 4), but there appears to be disagreement in the reconstruction of Tsum. While BRT and WA-PLS suggest an average Tsum of 20ºC, which is already surprisingly high, MAT indicates higher values (Fig. S4), which might be owed to the tendency of this method to overestimate parameters as it is more sensitive than the other two methods and has been shown in other studies to have a wider spread of estimates during glacial periods (Brewer *et al*., 2008; Sinopoli *et al*., 2019). On the other hand, precipitation reconstructions seem to be relatively in agreement with each other, suggesting dry conditions with PANN under 600mm. The results for this time period are also observed in other records and pollen-based reconstructions from southern Europe and the Iberian margin (e.g. Sanchez Goñi *et al*., 1999; Desprat *et al*., 2005; Brewer *et al*., 2008; Sinopoli *et al*., 2019; Leroy *et al*., 2023).

The transition from MIS 6 to MIS 5 is characterised by a rise in temperature and precipitation indicative of a gradually warmer and more humid climate. An increase in TANN is visible in all the three methods, from between 10–12 ºC during the glacial to 12–15 ºC at the beginning of MIS 5e, equivalent to the early Eemian (Fig. 4). This reflects the shift from steppic taxa to *Quercus* and other temperate vegetation (Fig. S1) as was also recorded in the marine records of MD952042 (Sanchez Goñi *et al*., 1999) and MD01-2447 (Desprat *et al*., 2007) from the Iberian Margin. This progressive rise is paralleled by the rise in $CO_2$ and $CH_4$ from Antarctica, and the decrease in $\delta^{18}O$ (Desprat *et al*., 2005; Voelker *et al*., 2010; Oliveira *et al*., 2016). However, this transition towards climatic amelioration is interrupted by a short-lived event of abrupt cooling and drying, observed already in MIS 19 and 11. These events have previously been observed throughout the interglacials MIS 19, 11 and 5 in records from the Iberian Margin (Sanchez Goñi *et al*., 1999; Desprat *et al*., 2007) and were considered to be events analogue to Younger Dryas events or Henrich-type events associated with the weakening of the AMOC during the deglaciation period.

While there are some differences between the methods in terms of the specific timing of the peak climatic optimum during the Eemian (something that is itself under particular debate in the literature, e.g. Sanchez Goñi et al., 1999), the reconstructions show that the highest temperatures (>15 ºC) and humidity (≥600 mm) occurred between 127–118 ka BP (Fig. 4, Tab. 4). This is coeval with the lightest isotopic $\delta^{18}O$ signature from the Iberian margin (Desprat *et al*., 2007) and highest sea-surface temperatures recorded in cores ODP 977 in the Alboran sea (Martrat *et al*., 2004). During this climatic optimum, Tsum and Twin values peak with values higher than MIS 19 and 11 averaging >23 ºC and 10 ºC, respectively, indicating increased temperature during both winter and summer months (Fig. S4). These parameters indicate a more humid and warmer climate during the optimum of the Eemian than the present day, which corroborates the findings of several other studies in Europe (Guiot *et al*., 1989; Cheddadi *et al*. 1998; Sanchez Goñi *et al*., 1999; Desprat *et al*., 2007; Brewer *et al*., 2008; Leroy *et al*., 2023). For example, reconstructions from Lake Ohrid, La Grande Pile, Les Echets and Le Bouchet also show a thermal maximum around this time, between 127 and 118 ka, followed by cooling around 117 ka (Brewer *et al*., 2008; Sinopoli *et al*., 2019).

Our results also match the findings by Brewer *et al*. (2008), who identified a difference between northern and southern Europe, whereby records from higher latitudes experiences a sharp drop in temperatures and precipitation following the optimum whereas the climate remained more stable conditions over a longer period in the south. Our reconstructions for ODP 976, similarly to those from Lake Ohrid (Sinopoli *et al*., 2019) and Lago di Monticchio (Allen *et al*., 1999; Brewer *et al*., 2008), exhibit a gradual and continuous cooling trend without a sudden decrease in temperatures and precipitation following the Eemian optimum, suggesting an intermediate climate signal more similar to southern European sites than northern ones, and possibly corroborating the idea of a weak latitudinal gradient during this period. However, our results for Psum and Pwin show that there was still strong seasonality during the Eemian climate optimum at least in the Western Mediterranean, reflected more by precipitation parameters than temperature (Fig. S4). During this period, our reconstructions show that, while the climate was overall wetter than the glacial of MIS 6 (as well as the latter parts of the Eemian), the climatic optimum was characterised by very dry summers and contrastingly wetter winters. This might be linked with a strong Mediterranean climate during this time around the Alboran Sea, as previously suggested by Sanchez Goñi *et al*. (1999) for the Iberian Margin.

The tail end of the optimum is characterised by a decrease in temperature and a rise in precipitation, visible across all three methods, in agreement with other European records (Guiot *et al*., 1989; Brewer *et al*., 2008; Sinopoli *et al*., 2019). Throughout the rest of the interglacial, several fluctuations are observed between cool and warm periods, also observed in other southern-European records, with counterparts in Atlantic $\delta^{18}O$ records (Sanchez Goñi *et al*., 1999; Desprat *et al*., 2007; Sinopoli *et al*., 2019). Specifically, these occurred around 115 ka BP (Melisey I), 105 ka BP (St. Germain Ib) and around 87 ka BP (Melisey II), events which are characterised by colonisation by *Cedrus* and steppic vegetation; these are alternated with temperate phases St. Germain Ia and





Ic, and St Germain II, during which heathlands and deciduous and Mediterranean forests expanded again (Sanchez
Goñi *et al*., 1999). These events correlate well with the first Dansgaard-Oeschger events (Dansgaard *et al*., 1993),
DO-25, 24 and 23 described by Masson-Delmotte *et al*. (2005). During this period of variability, our parameters
suggest a progressive rise in precipitation and a slow decline in temperature throughout MIS 5c and the rest of the
interglacial, consistent with climatic reconstructions from the Mediterranean such as Lake Ohrid (Sinopoli *et al*.,
2019) and Lago di Monticchio (Brewer *et al*., 2008; Sinopoli *et al*., 2019), as well as records from the Iberian
margin (Sanchez Goñi *et al*., 1999; Desprat *et al*., 2007) and eastern Mediterranean (Leroy *et al*., 2023), showing
similar trends throughout the Mediterranean. During MIS 5b, a notable drop in PANN is observed around 89–86
ka BP, alongside a moderate rise in TANN. During substage 5a, both parameters decrease further, consistent with
glacial inception of MIS 4 (Fig. 4).

**Table 4 - Summary of results of the pollen-based climatic reconstructions for MIS 6–5**

| Interval | Age (ka BP) | Summary |
|---|---|---|
| **MIS 5a and b cooling** | 98–80 | Drop in precipitation but a smaller rise in temperature around 89–86 ka BP. Parameters show a consistent decline in temperature during MIS 5a consistent with glacial inception moving towards MIS 4. |
| **MIS 5c and d warm events** | 116–98 | Progressive rise in precipitation and a slow decline in temperature during the rest of the interglacial. |
| **Eemian (5e)** | 128–116 | Highest temperatures (~15 ºC) and humidity (≥600 mm) observed between 127–118 ka BP. |
| **MIS 6/5 transition** | 133–128 | Rise in temperature visible in all three methods. Temperature increases from 10–12 ºC during the glacial to 12–15 ºC at the onset of MIS 5. |


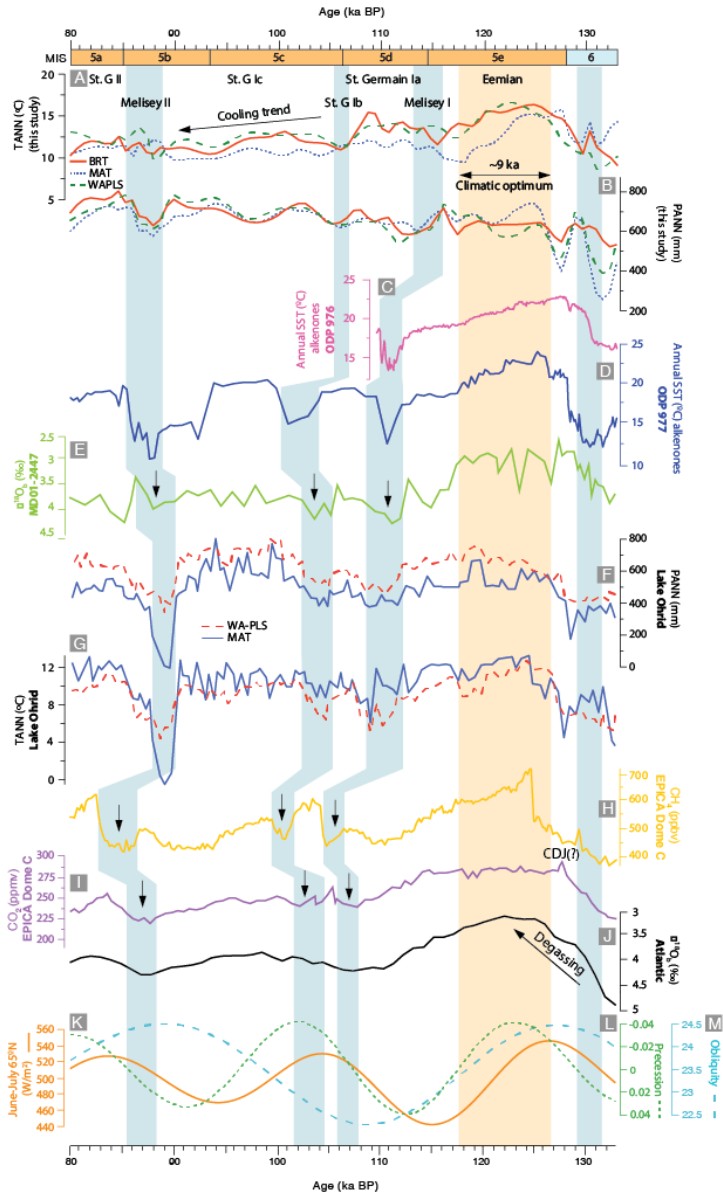

**Figure 4 – Comparison of the pollen-based quantitative reconstructions from ODP976 for MIS 5, (A)**
**TANN and (B) PANN (BRT=red solid line; MAT=blue dotted line; WA-PLS=green dashed line), with**
**other regional and global proxies: (C) Alkenone SSTs from ODP976 (Martrat et al., 2014); (D) Alkenone**
**SSTs from ODP977 (Martrat et al., 2004); (E) Benthic $\delta^{18}$O from MD01-2447 (Desprat et al., 2007); (F)**
**PANN and (G) TANN from Lake Ohrid derived through MAT and WAPLS (Sinopoli et al., 2019); (H)**
**Methane (CH$_4$) atmospheric concentrations (Loulergue et al., 2008) and (I) CO$_2$ atmospheric**
**concentrations from Antarctic EPICA Dome C ice cores (Nehrbass-Ahles et al., 2020); (J) Atlantic $\delta^{18}$O**
**(Voelker et al., 2010); (K) Summer insolation (Laskar et al., 2004); (L) Precession index and (M)**
**Obliquity curve (Berger and Loutre, 1991). Orange band indicates the period encompassing the climatic**
**optimum, and the blue bands highlight major millennial-scale climatic events.**



*4.2.4 MIS 2–MIS1 (21 ka BP–present day)*
*Last Glacial Maximum to HE-1*
During the Last Glacial Maximum (LGM), around 21–17.5 ka BP, MAT and WA-PLS suggest peculiarly high
TANN and Twin values, ranging between 12–15 ºC and 0–10ºC, respectively (Fig. 5, Tab. 5). MAT also suggests
drastically higher Tsum values during this period when compared with BRT and WA-PLS. This may once again
be due to the tendency of MAT to overestimate parameters during glacial periods, and is also linked to the inferior
reliability of WA-PLS when compared to the newer method BRT. Combourieu-Nebout *et al*. (2009) also noticed
that their MAT reconstruction for the end of the LGM were higher than expected and closer to present-day levels,
as it appears in the reconstruction methods of this current study. This discrepancy to the possible lack of good
present-day analogues for the cedar/heath pollen association which is dominant in the pollen record at the end of
the LGM (Combourieu-Nebout *et al*., 2009). In contrast, BRT suggests relatively lower annual and seasonal
temperatures than the other two methods for the LGM period, which is more in line with previous interpretations
made by Combourieu-Nebout *et al*. (2009) on the basis the ODP Site 976 pollen record during this period. In their
study, TANN reconstructions suggested anomalies around -5ºC and Twin between -10ºC and -15ºC. Overall,
precipitation during this period appears to be consistently low across all three methods, with PANN values
remaining below 600 mm/yr, indicating a dry climate in agreement with the previous study on the ODP 976 core
by Combourieu-Nebout *et al*. (2009), as well as the PANN reconstruction for the Padul record (Fig. 5) which
shows a period of low precipitation patterns between 20 and 16 ka BP consistent depleted $\delta_{DC31}$ values (Camuera
*et al* 2018, 2019, 2022; García-Alix *et al* 2021)
Between 17 and 15 ka BP, a drastic fall in temperature and precipitation is observed (Fig. 5). This change has
been previously attributed to the Oldest Dryas event in the south-western Mediterranean, consistent with Heinrich
Event 1 observed in several other marine and terrestrial records in the Mediterranean (Pons and Reille, 1988;
Watts *et al*., 1996; Combourieu-Nebout *et al*., 1998, 2002; Allen *et al*., 2002; Peñalba *et al*., 1997; Turon *et al*.,
2003; Naughton *et al*., 2007; Fletcher and Sanchez Goñi, 2008; Bordon *et al*., 2009) and has been interpreted as
increased dryness over the Alboran Sea (Combourieu *et al*., 2009). Our climatic reconstructions suggest minimum
temperatures with Twin values of -5–0ºC across all methods, and annual and seasonal precipitation values similar
to the late Pleniglacial with a minimum of ~300 mm shown by the MAT method. This event has a counterpart in
marine records for alkenone-derived SSTs from ODP Site 976 (Martrat *et al*., 2014) and other Mediterranean sites
(Kallel *et al*., 1997; Rohling *et al*., 1998; Cacho *et al*., 2001; Combourieu-Nebout *et al*., 2002; Perez Folgado *et
al*., 2003; Camuera *et al*., 2021). Recent studies from the new Padul record found a similar pattern in their PANN
and TANN reconstructions (Camuera *et al*., 2022; Rodrigo-Gámiz *et al*., 2022), suggesting comparable conditions
over the Western Mediterranean during this period. This has also been corroborated by Ludwig *et al*., (2018)
through model simulations of PANN and TANN over the Iberian Peninsula, which indicated a drastic decline in
both parameters during HE-1.
*Lateglacial, beginning of MIS 1*
A rise in temperature and precipitation is observed between 14.7 and 12.5 ka BP, shown consistently by the three
reconstruction methods (Fig. 5, Tab. 5). Although this is not reflected as strongly by the precipitation parameters,
temperature reconstructions achieved particularly with BRT ad WA-PLS show two distinctive periods of
increased warmth centred around 14 and 13 ka BP, attributed respectively to the Bølling and Allerød (B-A) warm
interstadials (Zonneveld, 1996; Combourieu-Nebout *et al*., 2009; Dormoy *et al*., 2009; Camuera *et al* 2019, 2021;
Rodrigo-Gamiz *et al*., 2022). During these periods, Twin values ranging 0–6ºC and TANN values of 12–14ºC
(Fig. 5, Fig. S5). Precipitation reconstructions suggest similar seasonality to the present-day in the Mediterranean,
with wet winters and dry summers as evidenced by the increase in Pwin but relatively consistent Psum values (
Fig. S5). In comparison with the values reconstructed for the Holocene, temperatures during the B-A remain
slightly subdued (Fig. 5).
Between 12.5 and 11.7 ka BP, all three methods indicate a return to colder and drier conditions compared to
the B-A interstadial, related to the Younger Dryas event (YD). Twin values during the YD range from
approximately -2ºC to 3ºC, and TANN values range from 10ºC to 13ºC. Precipitation is also low across all three
methods, especially in PANN and Pwin, which decline from 700mm and 300m during the B-A to 500mm and
250mm, respectively, during the YD. These results are similar to those reconstructed by Combourieu-Nebout *et
al*. (2009) but are slightly higher than the values reconstructed by Dormoy *et al*. (2009). A comparably colder and
more arid climate compared to the B-A in this region was also observed by Camuera *et al*. (2021, 2022) and by
Rodrigo-Gamiz *et al*. (2022), although their values were slightly higher for both parameters perhaps indicating a
slight difference on land within the Iberian Peninsula compared to the conditions in the Alboran Sea at this time.
Overall, however, our results show similar timings, trends and amplitudes to what has so far been observed in
regional records from the Mediterranean and Iberian Margin, and global proxies such as $CH_4$ records from
Antarctica (Jouzel *et al.*, 2007; Nehrbass-Ahles *et al.*, 2020).





*Holocene*

The transition from the YD to the Holocene at 11.7 ka BP is marked by an increase in temperature and precipitation
parameters across all three methods (Fig. 5). TANN reaches similar levels to the present-day, and PANN reaches
values above 600mm/yr. Seasonal temperature parameters Twin and Tsum show consistently high values with
warmer summers and slightly cooler winters. There is a large difference between Psum and Pwin, indicating
seasonal variation in wetness which may be related to the onset of present-day altitudinal vegetation belts and
Mediterranean climate (Combourieu-Nebout *et al.*, 2009). This amelioration is coeval with the increase in SST
values from ODP 976 which show warming in marine environments as well as on land at the beginning of the
Holocene (Combourieu-Nebout *et al.*, 2002, 2009). This is also shown by alkenone and foraminiferal-based SST
records in the nearby core MD 95-2042 (Cacho *et al.*, 2001; Perez Folgado *et al.*, 2003) and the $\delta^{13}$C and $\delta^{18}$O
depletion in the MD 90-917 core in the Adriatic Sea (Siani *et al.*, 2013).
Maximum temperatures and precipitation in our reconstructions mark the optimum climatic conditions of the
Holocene between 9 and 7 ka BP, consistent with other studies in the Mediterranean (Bar-Matthews *et al.*, 1998;
Rossignol-Strick, 1999; Kotthoff *et al.*, 2008; Ramos-Román *et al.* 2018, Marriner *et al.*, 2022), as well as in
central Europe (Magny *et al.*, 2002; Martin *et al.*, 2020; Cartapanis *et al.*, 2022; d'Oliveira *et al.*, 2023). As shown
by our Pwin and Psum values, seasonality is strong during this period—winter precipitation increases significantly
(300 to 400 mm/yr) while summer precipitation reaches a minimum (around 50 mm/yr) suggesting strong seasonal
contrasts. In the early Holocene, Twin values range from approximately -0.85 ℃ to 5.81 ℃, while Tsum values
range from 19.15 ℃ to 23.59 ℃. These findings match the reconstructions by Dormoy *et al.* (2009) and Jalut *et*
*al.* (2009) who suggested that in the Western and Central Mediterranean, the climatic optimum of the Holocene
was characterised by hot and dry summers and wet and cool winters. This has also been corroborated by more
recent climatic reconstructions for Padul (Ramos-Román *et al.*, 2018; Rodrigo-Gamiz *et al.*, 2022). This contrasts
with results from Northern and Eastern Europe, where high year-round moisture and wet summers prevailed
(Rossignol-Strick, 1999; Bar-Matthews *et al.*, 1998), consistent with the east-west precipitation gradient observed
during the climatic optima of Holocene analogues.
The optimum is interrupted by a short-lived cooling event around 8.4–8.2 ka BP, observed in many other
global records (Von Grafenstein *et al.*, 1998; Mayewski *et al.*, 2004; Alley and Agustsdottir, 2005; Pross *et al.*,
2009; Marriner *et al.*, 2022). The reduction in our reconstructed parameters during the 8.2 ka event, particularly
the reduction in precipitation although not as much in temperature, can be explained by a reduction in North
Atlantic Deep Water (NADW) formation due to increased meltwater from the Laurentide lakes into the North
Atlantic (Barber *et al.*, 1999; Ellison *et al.*, 2006).
The reconstructions for PANN indicate a generally decreasing trend for the last 7 ka with good consensus
between methods (Fig. 6, Table 6). Meanwhile, TANN shows different amplitudes of change; while BRT and
WAPLS indicate an overall upwards trend in temperatures between 6–2 ka BP, MAT suggests a comparatively
more drastic decline. Short-term fluctuations previously identified by Combourieu-Nebout *et al.* (2009) and
Dormoy *et al* (2009) are also observed in our record around 6–5, 4.3 and 3.7 ka BP, which roughly correlate with
Bond events in the North Atlantic (Bond *et al.*, 1997, 2001).
However, the climatic reconstructions from 7 ka onwards must be interpreted cautiously due to the increasing
anthropogenic impact during this period. The decline in temperature and precipitation parameters, rather than
being a result of progressive cooling, might in fact be an artificial result of increase in semi-desert taxa such as
*Artemisia* and reduction in temperate and Mediterranean forest cover (Fig. S1) related to anthropogenic impact in
the form of clearing (Combourieu-Nebout *et al.*, 2009). However, several other reconstructions for this period in
this region (Camuera *et al.*, 2022; Rodrigo-Gamiz *et al*, 2022; Liu *et al.*, 2023) and in Western Mediterranean (Di
Rita *et al.*, 2022) suggest similar findings. Liu *et al.* (2023) proposed that the consistency of climate
reconstructions during this period signifies that the changes observed are a reflection of regional climate rather
than human activity in the form of agriculture or landscape modification and therefore should be considered as
such. On the other hand, during the past 2 ka all methods indicate a substantial rise in temperatures and further
decline in precipitation, most likely reflecting at this point the increasing human influence on vegetation
composition, especially during the post-industrial era (Ruddiman *et al.*, 2016).

**Table 5 - Summary of results of the pollen-based climatic reconstructions for MIS 2–1**

| Interval | Age (ka BP) | Summary |
|---|---|---|
| **Middle-Late Holocene** | 6.4-present | BRT and WA-PLS indicate an overall upwards trend in temperatures. MAT suggests a comparatively more drastic decline. |
| **Early-Middle Holocene climatic optimum** | 11.7-6.4 | Consistent rise in temperature and precipitation by all three reconstructions. Climatic optimum observed between 11 and 7 ka BP. All methods show a temperature rise above 13 ℃, peak in precipitation reaching >700 mm. Interrupted by a noteworthy cold and dry event around 8.2 ka BP. |



| Younger Dryas | 12.5-11.7 | Return to colder and drier conditions<br>Twin values during YD range from approximately -2ºC to 3ºC, and TANN values range from 10ºC to 13ºC.<br>Precipitation is low across all three methods. |
|---|---|---|
| Bølling-Allerød | 15-12.5 | Temperature reconstructions show two distinctive periods of increased warmth.<br>Attributed to Bølling and Allerød warm interstadials.<br>Twin values ranging 0–6ºC and TANN values of 12–14ºC. |
| H1-Oldest Dryas | 16-15 | Drastic fall in temperature and precipitation observed, related to Oldest Dryas (H1)<br>Climatic reconstructions suggest minimum temperatures with Twin values of -5–0ºC.<br>Annual and seasonal precipitation values similar to late Pleniglacial (~300 mm shown by MAT method). |
| MIS 2/1 transition | 21.2-15 | MAT and WA-PLS show high TANN ranging between 12–15 ºC.<br>PANN indicates a large range of 500–800 mm across the three methods.<br>Significant drop in temperature and precipitation during H1; annual temperatures fall to 10–12 ºC<br>Precipitation falls below 600 mm (minimum ~300 mm shown by MAT). |

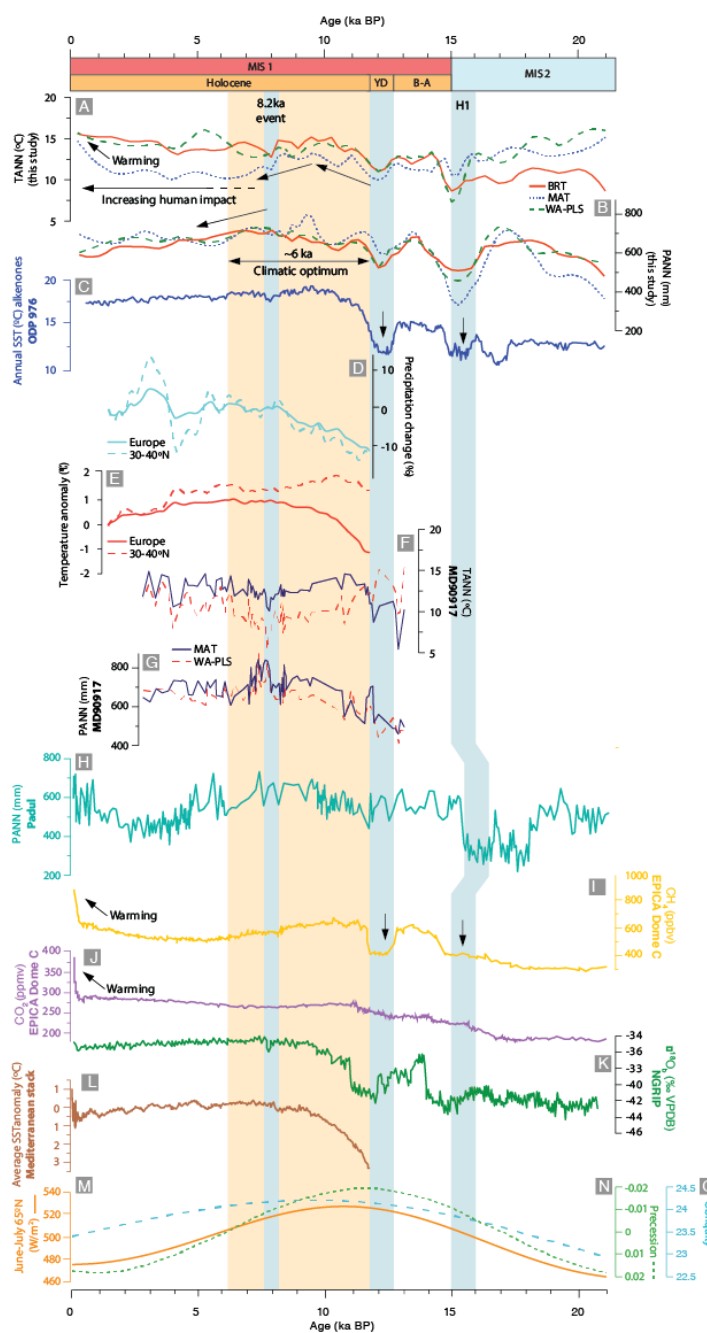

**Figure 5 – Comparison of the pollen-based quantitative reconstructions from ODP976 for MIS 1, (A)**
**TANN and (B) PANN (BRT=red solid line; MAT=blue dotted line; WA-PLS=green dashed line), with other**
**regional and global proxies: (C) Alkenone SSTs from ODP976 (Martrat et al., 2014); (D) Precipitation**
**change (%PANN) and (E) Temperature anomaly (TANN) for Europe derived from WAPLS (Herzschuh**
**et al., 2023); (F) PANN and (G) TANN obtained through quantitative pollen-based reconstructions using**
**MAT and WAPLS (Combourieu-Nebout et al., 2013); (H) Pollen-based quantitative reconstruction of**



**PANN from Padul derived using WAPLS (Camuera et al., 2023); (I) Methane (CH₄) atmospheric concentrations (Loulergue et al., 2008) and (J) $CO_2$ atmospheric concentrations from Antarctic EPICA Dome C ice cores (Nehrbass-Ahles et al., 2020); (K) NGRIP ice $\delta^{18}O$ (North Greenland Ice Core Project Members, 2004); (L) Average SST anomaly from Mediterranean stack (Marriner et al., 2022); (M) Summer insolation (Laskar et al., 2004); (N) Precession index and (O) Obliquity curve (Berger and Loutre, 1991). Orange band indicates the period encompassing the climatic optimum, and the blue bands highlight major millennial-scale climatic events.**

4.3 Interglacial analogues of the Holocene in the southwestern Mediterranean

The climate reconstructions show changes in temperature and precipitation in the Alboran Sea during MIS 19, 11, 5 and the Holocene (Fig. 6), which correlate with climatic changes observed in other regional and global proxies indicating that overall the reconstructed parameters are reasonable and reliable. Our reconstructions enable a valuable comparison of the structure and amplitude of millennial-scale climate variation during these periods in the southwestern Mediterranean.

Before delving into a discussion about how MIS 19, 11 and 5 compare climatically and their suitability as interglacial analogues of the Holocene, the implications of anthropogenic impact over the past 7 ka must be considered. The extent to which humans have altered the current interglacial and therefore what is considered 'natural' climate change has been subject of substantial debate over the past couple decades (Ruddiman, 2003, 2007; Ruddiman et al., 2016). This is particularly with regard to the origin of the $CO_2$ increase by 20 ppmv, as well as the rise in CH₄, during the late Holocene (Yin and Berger, 2015), believed to be a result of the clearing of forests and agricultures over the past 7 ka BP. Ruddiman (2003, 2007) hypothesised, under what is known as the early Anthropogenic hypothesis, that the rise in GHGs between 7 ka BP and the Industrial Era is not caused by natural sources but rather by human intervention in the form of forest clearance, livestock domestication and flooding of rice paddies (Ruddiman, 2003, 2007; Broecker and Stocker, 2006). The increase in GHGs resulting from preindustrial farming was enough to cause anomalous warming and prolonged the duration of the interglacial, whereas based on solar precession the Holocene would be expected to be nearing the end of its natural course (Yin and Berger et al., 2015). This hypothesis has significant implications on the reliability of comparisons between MIS 1 and the interglacial analogues, and leads to significantly different conclusions about the natural trajectory of the Holocene (Tzedakis, 2010). Some authors question the extent of anthropogenic impact on climate during the pre-Industrial period altogether, making the debate over the early Anthropogenic hypothesis somewhat irrelevant. Yin and Berger (2015) state that whether the hypothesis is right or wrong, the increase in $CO_2$ by 20 ppmv during the late Holocene is significantly smaller than the 120 ppmv released during the 20th and 21st centuries, and therefore the late Holocene can be considered natural enough to enable comparisons with other interglacials.

As shown in Figure 6, the MAT method suggests the latest warming trend has occurred over the last 2,000 years or so, while BRT and WA-PLS indicate a slower gradual warming over the past 4,000 years. This gradual increase coincides with the gradual increase in GHGs evidenced by the CH₄ and $CO_2$ EPICA records, and these also indicate that it is only in the most recent centuries that peak values are recorded, i.e. since the Industrial Era (Jouzel et al., 2007; Pol, 2010; Nehrbass-Ahles et al., 2020). While the importance of human forcing on climate is recognised, the idea that pre-Industrial activity represented a small enough contribution to GHG emissions is still entertained to allow comparisons between the Holocene and Pleistocene interglacials.

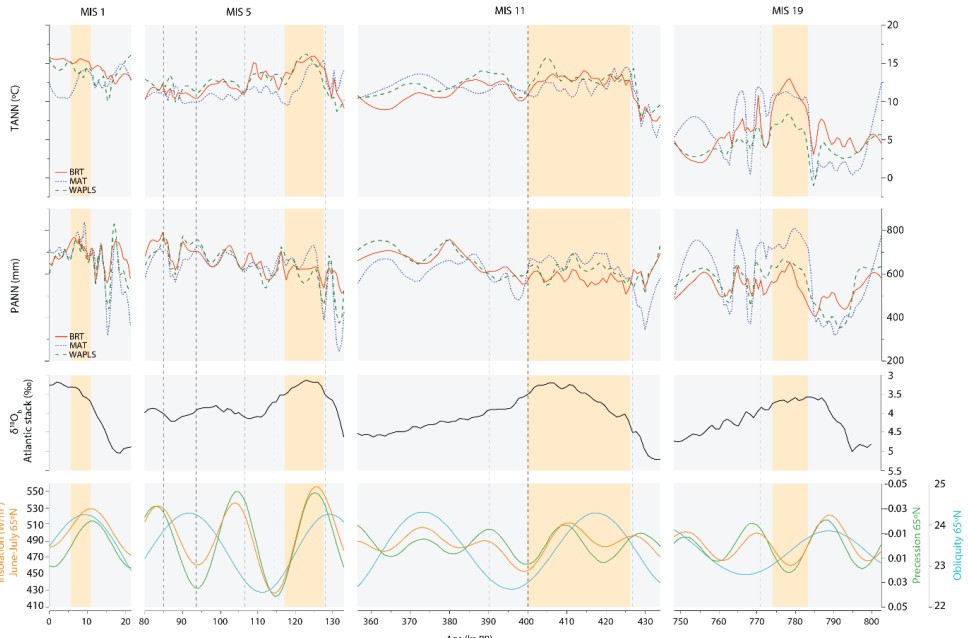

**Figure 6 - Comparison of the quantitative pollen-based reconstructions (TANN and PANN) from ODP976 for MIS 19, 11, 5 and 1, compared with the Atlantic δ¹⁸O stack by Voelker et al. (2010) and solar orbital patterns: Summer insolation (Laskar et al., 2004), Precession index and Obliquity curve (Berger and Loutre, 1991). Orange bars indicate the period encompassing the climatic optimum in each interglacial.**

The reconstructions for MIS 19 (Fig. 6) display the highest degree of variability throughout the interglacial, with high-amplitude fluctuations across all three methods between warm and colder substages. Generally, the models show a colder climate than the other interglacials (Fig. 6). These match the findings of other authors and it has been widely recognised that MIS 19 is colder than the interglacials after Termination V (Jouzel *et al*., 2007; Candy *et al*., 2014, 2024). When comparing to the EPICA records of MIS 19 to those of the other interglacials, the former shows lower concentrations of GHGs (Pol, 2010; Nehrbass-Ahles *et al*., 2020), supporting our findings of lower temperatures during this period. A colder climate than present during the climatic optimum of MIS 19c has been observed by Jouzel *et al*. (2007), who stated that this period was characterized by less pronounced warmth than interglacials MIS 5e, 7e, 9c, and 11c. Moreover, a main distinction between MIS 19 and the Holocene is that following the peak of MIS 19, temperatures decline relatively quickly, while during Holocene there is a short-lived decline in temperature, followed by a renewed increase and stabilisation during the Late Holocene (Candy *et al*., 2014). In general, while the solar forcing of MIS 19 might be more similar to MIS 1, the climatic structure of MIS 19 has little resemblance to MIS 1 when considering the duration of the sustained warmth during the pre-Industrial Holocene, at least in the region around the Alboran Sea.

MIS 11 differs from MIS 19 in the magnitude of temperature variations. It is also much longer than both MIS 19 and MIS 5, and indeed the Holocene, due to its unique antiphasing between insolation and obliquity (Ruddiman *et al*., 2007; Nomade *et al*. 2019; Tzedakis, 2010; Tzedakis *et al*., 2022). While MIS 11 exhibits warmer temperatures compared to MIS 19, it still shows some degree of variability as observed with its high- and moderate-intensity climatic variability events and climatic fluctuations during the optimum, like the OHO. Overall, however, it is significantly more stable than MIS 19. According to Candy *et al* (2014), if the early Anthropogenic hypothesis is not accepted, MIS 11c is a closer climatic analogue, which means that the current interglacial may last for over 50 ka (Loutre and Berger, 2003; McManus *et al*., 2003; Candy *et al*., 2014). If instead this hypothesis is accepted then MIS 19 and MIS 1 become more similar, meaning that the current interglacial would be close to its end if it weren't for anthropogenic forcing (Candy *et al*., 2014; Tzedakis, 2010). The key particularity of accepting MIS 11 as an analogue is that it is the only interglacial with a combination of elevated GHG concentrations and an extended duration. Considering that human activity is affecting the length of the Holocene (Tzedakis *et al*., 2012; IPCC 2022), this makes MIS 11c an important analogue for how the earth´s climatic system functions under extended interglacial conditions (Candy *et al*., 2014, 2024).



Similarly to MIS 11, MIS 5 is characterised by elevated greenhouse gas levels and high sea levels, although this interglacial has been criticised as an analogue by previous authors due to its high-amplitude fluctuations in solar forcing. The reconstructions for MIS 5, particularly for MIS 5e (the Eemian), suggest a significantly warmer climate regime compared with the other interglacial analogues. In terms of duration, MIS 5e is slightly shorter than MIS 19, but similarly to MIS 11 it exhibits more stable climatic conditions as also corroborated by the lower variation in SSTs in records form the Western Mediterranean (Martrat *et al.*, 2004). A warmer climate than other interglacial analogues and the Holocene (specifically, warmer than pre-Industrial levels) has been previously observed for the Eemian, for example at Padul (Camuera *et al.*, 2019), La Grande Pile (Guiot *et al* 1989; Brewer *et al*, 2008) and in the North Atlantic (Zhuravleva, 2018). On a global average, MIS 5e has been found to be the warmest interglacial of the past 800 kyr (Tzedakis *et al.*, 2022). When considering the factors together, i.e. significantly higher temperatures, short duration, and high-amplitude fluctuations in solar forcing, in the case of our reconstructions MIS 5 appears to be the least suitable analogue when compared with MIS 19 and 11.

Our high-resolution climatic reconstructions have demonstrated that in terms of magnitude of warmth, structure, stability and duration the interglacial analogues of the Holocene are, fundamentally, unique. Although they all are reoccurring events and share similar patterns such as the abrupt shifts from glacial to interglacial, the occurrence of climatic optimums soon after the transition, and cold events and Younger-Dryas-like events, the associated climate feedbacks in each interglacial produce very different climatic histories that are difficult to compare with the Holocene. As Candy *et al.* (2014) point out, there is no reason to expect that the climate of MIS 1 should naturally follow the pattern of MIS 11 or 19 or indeed MIS 5, despite the close similarities in insolation forcing, greenhouse gas concentration and temperatures. The study of past interglacials does not offer a direct blueprint for predicting the future evolution of the Holocene. However, these interglacial analogues are valuable for exploring the responses of the Earth's processes under different forcing factors which closely resemble the climate system during the Holocene. What emerges from the climatic reconstructions from ODP Site 976 and the close comparisons with global and regional records is that this site is extremely sensitive to global changes which in turn can be used to infer that the southwestern Mediterranean will be highly susceptible to future climate change and anthropogenic forcing.

## 5. Conclusion

This study has provided valuable insights into the climatic variations during MIS 19, 11, 5 and 1, within the southwestern Mediterranean region. Through pollen-based climatic reconstructions, we have established correlations between temperature and precipitation changes in our study area with those observed in other regional and global proxies, confirming the reliability of our findings. These reconstructions facilitate a comprehensive comparison of millennial-scale climate variations during these interglacials, shedding light on their unique climatic structures and amplitudes.

The reconstructions highlight a temperature increase from MIS 19 to the Holocene and distinct climatic characteristics of each interglacial period. MIS 19 exhibits high variability and colder temperatures compared to subsequent interglacials and the Holocene. Conversely, MIS 11 displays warmer temperatures and greater stability, offering an insight into interglacials of prolonged duration, crucial when considering that the anthropogenically-driven warming of the post-Industrial era might be artificially prolonging the current interglacial. Reconstructions for MIS 5 suggested overall warmer conditions, especially during the Eemian, but this higher temperature coupled with high-amplitude fluctuations in solar forcing makes it a less suitable Holocene analogue.

While past interglacials do not provide a straightforward blueprint for predicting the future evolution of MIS 1, they offer invaluable insights into Earth's responses to different forcing factors during periods with similar climatic conditions to the Holocene. The pollen-based climatic reconstructions for MIS 19, 11 and 5 serve as crucial benchmarks for understanding the sensitivity of the southwestern Mediterranean to global changes, and underscore the importance of mitigating climate change in this region.

## Competing interests

At least one of the (co-)authors is a member of the editorial board of Climate of the Past.

## Acknowledgments

We sincerely appreciate the financial support from the ANR project Neandroots (Agence Nationale de la Recherche, project No. ANR-19-CE27-0011-01), the Muséum national d'Histoire naturelle (MNHN), and the Centre National de la Recherche Scientifique (CNRS). apThanks to the ISEM, the Institut des Sciences de l'Évolution de Montpellier, UMR CNRS 5554 ISEM (Université de Montpellier) for hosting D. Sassoon on multiple occasions for training on transfer functions. Special thanks to Léa d'Oliveira for the assistance with the transfer function models and for her help troubleshooting the scripts. This is an ISEM contribution number XXX (tbc).



**Supplementary data**
Supplementary data to this article can be found online at https://data.mendeley.com/datasets/m4kzgwk6b9/1

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
