# Peer review of "Pollen-based climatic reconstructions for the interglacial analogues of MIS 1 (MIS 19, 11 and 5) in the Southwestern Mediterranean: insights from ODP Site 976"

_EGUsphere, 2024_

## Author Response (AR2)

**Authors' response:**

We thank the reviewers for their valuable comments, which will help improve the clarity and robustness of our manuscript. Below, we provide a detailed point-by-point response to each comment.

Anonymous Referee #1:

Review/Discussion of "Pollen-based climatic reconstructions for the interglacial analogues of MIS 1 (MIS 19, 11 and 5) in the Southwestern Mediterranean: insights from ODP Site 976"

1. Does the paper address relevant scientific questions within the scope of CP? YES
2. Does the paper present novel concepts, ideas, tools, or data? YES
3. Are substantial conclusions reached? YES, though there is room for improving the discussions and conclusions
4. Are the scientific methods and assumptions valid and clearly outlined? YES
5. Are the results sufficient to support the interpretations and conclusions? YES
6. Is the description of experiments and calculations sufficiently complete and precise to allow their reproduction by fellow scientists (traceability of results)? YES, but could be improved in some aspects
7. Do the authors give proper credit to related work and clearly indicate their own new/original contribution? YES
8. Does the title clearly reflect the contents of the paper? YES
9. Does the abstract provide a concise and complete summary?
10. Is the overall presentation well structured and clear? YES
11. Is the language fluent and precise? YES, but compare some remarks below
12. Are mathematical formulae, symbols, abbreviations, and units correctly defined and used? YES
13. Should any parts of the paper (text, formulae, figures, tables) be clarified, reduced, combined, or eliminated? In some cases, there is room for clarifying/improving
14. Are the number and quality of references appropriate? YES, see below
15. Is the amount and quality of supplementary material appropriate? YES, but some of the mentioned aspects for which additional information could help (e.g. age model) could be handled via supplementary material

This is an interesting and generally well written and illustrated article which should be printed in climate of the past. The approach to compare the MIS 1 with its putative "best analogues" in terms of orbital cycles at the same spot from a pollen perspective is to be appreciated. It is also nice that three different methods of pollen-based quantitative are used and their results are compared. The reasons for the differences in the pollen-based reconstructions may perhaps be worth a more detailed discussion though (there is a theoretical comparison based on the reliability tests in section 4.1, but for the readers it might be interesting to get an idea how the method differences influence the results shown later for the record).

Response: Thank you for your valuable suggestion. We agree that a more detailed discussion of the methods could be beneficial and we intended to expand on this in our original draft.

However, we chose to keep the discussion on methods concise to maintain the focus of the discussion on the key findings, particularly the more interesting aspects of the four interglacials. In an effort to reduce the length of the manuscript, we opted to dedicate a specific section to the methods, while keeping the rest of the discussion focused on interpreting the results and their broader implications.

In addition to some smaller details mentioned below, one weakness is to me that the discussion does not go into more detail concerning the processes which in addition to the orbital cycles could cause the similarities and dissimilarities between the four interglacials and the preceding transitional phases (for example ice sheet development and related fresh water pulses; solar cycles; ecological conditions before the transitions). Also the aspect of anthropogenic influence during MIS 1 could be more precisely discussed (but I may misunderstand some statements made in the text, see below). On the other hand, the results presented here are probably an excellent base for further studies which could include focus on the driving factors.

Response: Although we agree that a more detailed discussion of the processes beyond orbital cycles would be interesting, but our aim in this paper was to focus on identifying and analysing the key patterns of similarities and dissimilarities between the four interglacials, rather than delving into all potential drivers and especially those prior to the transitions. We feel that a broader discussion of the additional processes would detract from our primary objectives given the current length and focus of the paper, though they undoubtedly represent promising avenues for future research. Regarding the anthropogenic influence during MIS 1, we have clarified the relevant statements to ensure they are more precise and accurately reflect our discussion of this aspect (see our response about this below).

Detailed remarks and suggestions:

Line 63:

The wording "MIS5… e… known as the Eemian" seems a little bit problematic, I suggest to use a wording such as "equivalent to" or "corresponding to".

Response: We agree with the suggestion and have revised the wording to "equivalent to the Eemian."

Line 139:

"The Alboran Sea is dominated…" While this is understandable, the wording seems a little odd to me.

Response: We have rephrased the sentence for clarity to: "Circulation in the Alboran Sea is influenced by the exchange of waters at the Strait of Gibraltar whereby low-salinity waters from the Atlantic…"

Line 179:

Is it still the age model from Combourieu-Nebout et al. 2009, with calibrations based on Bard et al./Stuiver et al. (1998)? Newer calibrations may not necessarily be "better", but knowing which calibration was used may be important when comparing with other data.

Response: We have updated the section to make it clear what the original calibration in Combourieu-Nebout et al. (2009) is based on: "based on calibrations by Bard et al. (1998), Stuiver and Reimer (1993) and Stuiver et al. (1998)"

Line 191 and elsewhere: I am not a native speaker, so I may be wrong, but in some cases I would add a "The" in front of the mentioned techniques, e.g. "The MAT and the WA-PLS have…"

Response: Both options (with and without 'the') are correct based on previous papers that implemented these papers. We have added 'the' where appropriate.

Line 191 and following:

Here, a lot of citations are given of which some are only once more or even nowhere else used in the publication. On the other hand, there is no citation given after the second sentence ("The results are often well-supported by other Mediterranean records…"). I suggest to reduce some citations (particularly those not used elsewhere) above and instead name some examples in which other proxies supported the MAT/WA-PLS results.

Response: We have reduce the number of citations to focus on key relevant studies that support the use of these methods in this sentence, and have added prominent examples where MAT and WA-PLS results are used in other Mediterranean proxy records in the second sentence. See lines 197-208 in marked up version.

Line 199 and perhaps elsewhere: May be nitpicking and rather a writing style aspect, but can a technique "use" things/approaches? Things can be used while applying a technique.

Response: We agree that the phrasing could be more precise. We have revised this to the following phrasing: "The MAT technique involves applying information from…"

Line 257 and following: I am not acquainted with the WA-PLS. In any case, also for the other methods, I wondered if it could also be possible to control the results by applying the techniques to datasets for which instrumental data is present. Maybe studies in that direction exist and can be cited?

Response: We don't have enough pollen data with sufficiently high resolution to compare with instrumental data. What is often used in other studies of this sort, and indeed is what we also included, is the correlation coefficient and the RMSE. With regard to the WA-PLS, other studies also show that it is less reliable than MAT and BRT (see for example d'Oliveira et al., 2023 and Dugerdil et al., 2021.) We have cited these papers in our manuscript for clarity.

Line 448 and following: I suggest to add here (or later in 4.3?) if the transitions to MIS19/11/5 have YD/H1-like events in other European records or only in the Mediterranean region. In Central/Northern Europe or the eastern Mediterranean long records (TP/Ohrid), at least the transition to MIS5e seems to lack a "real cold event" (as shown for Ohrid in Fig. 4/Sinopoli et al. 2019).

Response: This is a good point and looking through the literature the YD/H1-like events seem to occur mainly in the Mediterranean in marine core sites or Tenaghi Philippon, although this may be a result of these records receiving more regional signals compared to terrestrial records in mainland Europe. Still, looking at the PANN and TANN records for Lake Ohrid in

figure 4, albeit with a time-lag, there is a short-lived change to drier and colder condtions around 128 ka BP which could be linked to one of these cold events. Given the current focus and length of the discussion, we have made a short addition to the discussion in lines 471-475 and reflected this in figure 4.

Line 532 and Fig. 5:

Looking at publications on ice core records from Greenland and on records of the past glacial-Holocene transition from the Mediterranean, there are very different ways the H1 stadial/H1 event and the Oldest Dryas are handled. Rasmussen et al. (2014) dismiss the term Oldest Dryas. What seems reasonable to me based on quite recent results from the Mediterranean, including datasets shown in figure 5, is to regard the interval around 18 to 15 ka as the H1 stadial (quite equivalent to GS-2.1a in the ice core data), which comprises particularly cold events around 16.5 and 15.3 ka. While the event at 16.5 is weaker in the ODP976 pollen-based record, it seems to be stronger in other records and also in the ODP976 alkenone record. Thus I suggest not mark the interval between 16 and 15 ka as H1, but find a different solution. In the text, it may be worth to mention the different handling of the term Oldest Dryas if it shall be used at all.

Response: Given the problematic use of the term Oldest Dryas in previous literature, and the passing mention in our paper, we have decided to omit this term as it is not essential to our discussion.

Line 657 and following:

"Some authors question… somewhat irrelevant." I do not understand what is implied here and in the following paragraph (663-669). I guess what is meant is that your data supports the idea that anthropogenic influence contributed to warming particularly for the past 2000 years, but may have played a role also earlier. As mentioned, I am not a native speaker, so it may be my limited English abilities, but particularly the sentence starting in line 667 is not clear to me.

Response: We will streamline this section by firstly removing the sentence in line 657 (now 690 in marked up version), and revising the following section. In this section, we are simply introducing the concept of the anthropogenic warming hypothesis during the last 4000 years, and entertaining the idea that this might be plausible based on the slight increase in our temperature reconstructions, though we are not stating that our data supports it necessarily. We have tried to clarify this in the text between lines 690-703 in marked up version.

Line 751:

"valuable insights": It should be self-explanatory that the insights are valuable.

Response: we agree and we have removed this phrasing.

Table 4: At least in the online version, the Eemian summary is in a different text format. Maybe the temperature drop during the MIS6 and MIS5 transition (even if not that clear) is worth mentioning in the table?

Response: We have adjusted the formatting of Table 4 and included a brief mention of the temperature drop during the MIS 6 to MIS 5 transition, even though the signal is not as prominent.

General remark to the figures:

Depending on the resolution and size in the final article, the text size could be slightly larger. The yellow curves (EPICA Dome) may be difficult to see if printed on certain paper types or depending on the used screen.

Fig. 6: The texts could be particularly difficult to read if the figure is not printed over at least have a page, and the dotted lines are difficult to see. I think it may also be interested to see the graphs for the four transitions shown vertically so that one can more easily compare the timing, but perhaps this is not possible if all curves are shown…

Response: We have increased the text size in the figures for better readability in the printed version. Additionally, we have improved the visibility of the yellow curves and text as well as the thickness of the dotted lines.

Anonymous Referee #2:

Sassoon et al. have carried out pollen-based climate reconstructions for past interglacial periods that may serve as analogues for the Holocene to better understand the natural climate variability in the western Mediterranean region. Specifically, they used previously published pollen data from ODP Site 976 to reconstruct temperature and precipitation variability across Marine Isotope Stages (MIS) 19, 11, 5 and 1, employing three different techniques, i.e., the Modern Analogue Technique (MAT), the Weighted Average Partial Least Squares regression (WA-PLS), and the Boosted Regression Trees (BRT). Despite the differences in the climate estimates provided by each method, the overall picture is that each of these interglacials exhibits distinct characteristics. For instance, MIS 19 is cooler and marked by higher amplitude fluctuations than the other interglacials, temperature remained at higher levels for a longer period in MIS 11 compared to MIS 19 and MIS 5, and the highest temperatures were recorded during MIS 5. These results are in good agreement with regional and global climate records, suggesting that ODP Site 976 is a sensitive recorder of global climate change.

To my knowledge, this is the only available comparison of pollen-based climate reconstructions for these critical interglacials from a single site. The findings are certainly of interest, and the manuscript is well-written and clear. I provide below some (minor to moderate) comments that should be addressed prior to publication.

MIS 19:

The comparison of the data from ODP 976 with other regional and global records presented in Figure 2 is not straightforward. For example, peak $CO_2$ and $CH_4$ values from Antarctica during MIS 19 align with the coldest conditions during MIS 20 at Site 976. This is most likely related to age model uncertainties of each individual record. However, this issue has to be clearly addressed in the manuscript. The authors should also use the most recent age models for each of the records used in this comparison (e.g., Bereiter et al. 2015 for the EDC ice core data etc). Due to these uncertainties, the vertical color bars highlighting warm and cold intervals are misleading. For example, the blue bar at around 785 ka aligns a cold period at Site 976 with warm intervals in other records. Additionally, the yellow bar (warmest period in Site 976) doesn't align with the warmest periods in other records.

Also, how do the authors explain the discrepancy between their climate reconstructions and the data from Sulmona basin during MIS 20? The climate regime between these two sites appears very different for an extended period of several thousand of years.

Response: As stated by the reviewer, the discrepancies between models are most likely a result of age model uncertainties and we have clarified in the text (see methodology) that records have different age models derived from different proxies, so figures might appear misaligned as a result. We have double checked that the records (EDC ice core, etc.) and their relative age models are the most up to date (for the EDC core, for example, we use the chronology by Nehrbass-Ahles et al. (2020) which is the most recent). The blue bars during the beginning of MIS19 was tweaked to reduce any misinterpretations, although we would like to note (as this also comes up later in the comments) that the blue and yellow bars are used to indicate broad changes and aid interpretations, and are not being use to accurately link records.

With regards to the d18O record from Sulmona, this is most likely a difference in the age model. We thank the reviewer for pointing this mismatch, as we have noticed that in the paper by Giaccio et al (2015) their climatic optimum for MIS19 occurs earlier, which is not reflected in our figure. We have adjusted the blue-coloured bands to clarify the discrepancy/lag between our record and Sulmona and have updated the Sulmona curve with data provided directly by B. Giaccio.

MIS 11:

I have concerns regarding the reliability of the precipitation reconstructions. The glacial intervals MIS 12 and 10 show very high precipitation values, which is difficult to reconcile. The BRT results even indicate that the lowest precipitation occurred during MIS 11c (in contrast to what is stated in line 363). The authors need to clearly discuss the limitations of the methods applied as well as the uncertainties of their reconstructed values. Although the authors mention that the temperature reconstructions are more reliable than those for precipitation in section 4.1, this should be also elaborated here for clarity. Additionally, a comparison with other proxy data would be very useful. For example, how do the pollen-based reconstructions compare to the dD trends from Tenaghi Philippon?

As also mentioned above, the colored bars in Figure 3 require further consideration. For instance, I don't see any cooling at c. 408 and c. 390 kyrs in the Tann reconstructions from Site 976. Also, the cooling at the end of MIS 12 seems to align with a peak in SSTs at the same record and high temperatures at Tenaghi Philippon.

Response: While we acknowledge that the precipitation estimates, particularly for MIS 12 and 10, may appear challenging to reconcile, our aim was to focus on the overall patterns (mainly of the interglacial MIS 11) rather than delve too deeply into methodological uncertainties. We have already indicated in section 4.1 that temperature reconstructions are more reliable, and we believe this addresses the potential uncertainties without detracting from the broader implications of the results. Where we have clarified any glaring uncertainties, but given the current focus and length of the discussion, we chose to prioritize the more significant insights from the key findings of the interglacials. This applies to more detailed comparisons of other proxy data as well – this paper aims to discuss the reconstructions of four interglacials and there is so much we can include while keeping the manuscript succinct. In this case too, we have made sure that the blue bars connecting proxy records are better aligned.

MIS 5:

As for MIS 19 and 11, the authors need to carefully consider the alignments of the individual cold/warm intervals at Site 976 with other records. I can't help noticing that the blue bar during the end of MIS 6, which represents a cold period, actually highlights a peak in the Tann reconstructions.

Response: We have realigned the coloured bars in figure 4, but once again the blue and yellow bars are intended to indicate broad patterns of change and to aid in interpretation, rather than to provide an exact match between records. These bars are not designed to align with every proxy, but rather to serve as visual guides for identifying general trends.

Holocene:

The reconstructed data don't show a cooling and drying trend for the Late Holocene in contrast to other pollen-based reconstructions. I would like to see a more rigorous discussion on why this site differs from other Holocene records. Moreover, the authors should elaborate on how the three applied methods perform and which one seems more or less sensitive to the anthropogenic signal during the Holocene.

Response: Although there are some differences with other pollen-based reconstructions, we argue that the overall trend of our results is similar to other studies, especially that from Padul (Camuera et al., 2023). The potential differences seen between our reconstructed records and other reconstructions for the Late Holocene are due, first and foremost, to a lack of top samples in our record meaning that our results might have less constraints for this period. Also, the differences may well be attributed to a combination of site-specific factors. As we discussed in this current manuscript and our previous paper (Sassoon et al, 2023), the southwestern Mediterranean in general) tends to have a less pronounced response to terrestrial drying, especially when compared to other Mediterranean records more directly influenced by continental conditions. Therefore, the pollen assemblages in this region may reflect more stable local conditions, likely contributing to the difference in Late Holocene trends. We have repeated these points in our discussion – see lines 632-638 in the marked up file.

Regarding the anthropogenic signal possibly affecting our reconstructions, while we suggest that the overall trend of warming throughout the last few thousand years could be affected by humans based on comparisons with other studies, we don't record a strong enough pollen signal from anthropogenic taxa like Cerealia, Rumex, Plantago lanceoloata or Brassicaceae to see a substantial difference between the climatic reconstructions. Similarly to our response above, the differences between methods are also likely to be due to the lack of samples constraining the models in the Late Holocene. As we did not clarify this in our original discussion, we have included a comment about this – see lines 643-646 in marked up file.

Other comments:

Lines 191-196: There is no need for such a long list of cited papers; some of them also don't use pollen-based reconstructions (e.g., Ardenghi et al. 2019). Please consider citing only few key relevant papers for this work.

Line 293: Correct typo (Twhis to This).

Line 394: Unclear what the sentence 'synchronous cooling on land sea' means. The pollen data used in this study can only provide information about terrestrial climates.

Like 713: 'like the OHO' is unclear.

Response: We have corrected the typographical errors in line 293 and 713, and clarified the sentence in line 394. We also shortened the list of citations in lines 191-196 to focus on the most relevant studies for pollen-based reconstructions.